# The Ca²⁺-activated K⁺ current of human sperm is mediated by Slo3

Christoph Brenker[1], Yu Zhou[2], Astrid Müller[1], Fabio Andres Echeverry[1], Christian Trötschel[3], Ansgar Poetsch[3], Xiao-Ming Xia[2], Wolfgang Bönigk[1], Christopher J Lingle[2]*, U Benjamin Kaupp[1], Timo Strünker[1]*

[1]Department of Molecular Sensory Systems, Center of Advanced European Studies and Research, Bonn, Germany; [2]Department of Anesthesiology, Washington University School of Medicine, St. Louis, United States; [3]Lehrstuhl Biochemie der Pflanzen, Ruhr-Universität Bochum, Bochum, Germany

**Abstract** Sperm are equipped with a unique set of ion channels that orchestrate fertilization. In mouse sperm, the principal K⁺ current ($I_{KSper}$) is carried by the Slo3 channel, which sets the membrane potential ($V_m$) in a strongly $pH_i$-dependent manner. Here, we show that $I_{KSper}$ in human sperm is activated weakly by $pH_i$ and more strongly by $Ca^{2+}$. Correspondingly, $V_m$ is strongly regulated by $Ca^{2+}$ and less so by $pH_i$. We find that inhibitors of Slo3 suppress human $I_{KSper}$, and we identify the Slo3 protein in the flagellum of human sperm. Moreover, heterologously expressed human Slo3, but not mouse Slo3, is activated by $Ca^{2+}$ rather than by alkaline $pH_i$; current–voltage relations of human Slo3 and human $I_{KSper}$ are similar. We conclude that Slo3 represents the principal K⁺ channel in human sperm that carries the $Ca^{2+}$-activated $I_{KSper}$ current. We propose that, in human sperm, the progesterone-evoked $Ca^{2+}$ influx carried by voltage-gated CatSper channels is limited by $Ca^{2+}$-controlled hyperpolarization via Slo3.

*For correspondence: clingle@ morpheus.wustl.edu (CJL); timo. struenker@caesar.de (TS)

**Competing interests:** The authors declare that no competing interests exist.

**Reviewing editor**: Richard Aldrich, The University of Texas at Austin, United States

## Introduction

Mammalian sperm fulfil several demanding functions during fertilization: sperm track down the oocyte presumably by chemotaxis, rheotaxis, or thermotaxis (*Bahat et al., 2003*; *Eisenbach and Giojalas, 2006*; *Kaupp et al., 2008*; *Miki and Clapham, 2013*). Moreover, sperm break through the oocyte's vestments by hyperactivation and acrosomal exocytosis (*Florman et al., 2008*; *Ho and Suarez, 2001*). Sperm acquire these skills inside the female genital tract during a maturation process called capacitation (*Fraser, 2010*). Navigation, capacitation, hyperactivation, and acrosomal exocytosis are controlled by changes in intracellular pH ($pH_i$), membrane voltage ($V_m$), and intracellular $Ca^{2+}$ concentration ($[Ca^{2+}]_i$) (*Ho and Suarez, 2001*; *Eisenbach and Giojalas, 2006*; *Florman et al., 2008*; *Kaupp et al., 2008*; *Publicover et al., 2008*). These cellular responses are orchestrated by a set of unique ion channels (*Darszon et al., 2011*; *Lishko et al., 2012*; *Santi et al., 2013*).

A picture has emerged that the inventory and control of ion channels in mouse and human sperm are surprisingly different. For example, human but not mouse sperm harbor functional proton Hv1 channels (*Lishko et al., 2010*); purinergic P2X channels are functional in mouse (*Navarro et al., 2011*), but not in human sperm (*Brenker et al., 2012*); the sperm-specific CatSper (cation channel of sperm) $Ca^{2+}$ channel is activated by progesterone in humans (*Lishko et al., 2011*; *Strünker et al., 2011*), but not in mouse (*Lishko et al., 2011*). These different channel inventories and different mechanisms of channel activation might reflect adaptations to species-specific challenges encountered by sperm in the female genital tract.

In mouse sperm, an alkaline-activated K⁺ current, called $I_{KSper}$, is a critical determinant of $V_m$ and, thereby, controls other $V_m$-dependent channels (*Navarro et al., 2007*). Mouse $I_{KSper}$ is carried by the

**eLife digest** A sperm that has been ejaculated into the female reproductive tract must complete a number of tasks to pass on its genes to the next generation. First it must travel along a meandering route to encounter an egg, before pushing through a jelly-like coating that surrounds the egg and then, finally, fusing with the egg's surface membrane. In order to complete these steps and fertilise the egg, a sperm must undergo a process called 'capacitation'. This process, and a variety of other sperm functions, involves the controlled flux of positive ions into and out of the sperm via specific ion channels that are located in the cell membrane.

The properties of the ion channels that allow protons and calcium ions to move into and out of human sperm are well understood, but less is known about the channels that control the movement of potassium ions. In mice, a channel called Slo3 allows potassium ions to flow out of the sperm and makes the membrane voltage of these cells more negative. Also, in mice, this channel is essential for the sperm to function correctly, and for fertilization. However, in humans, it is unclear if the Slo3 channel is present in sperm and if it performs the same role.

Now, Brenker et al. have shown that the flow of potassium ions out of human sperm occurs via the Slo3 channel, and that human Slo3 is responsible for setting the membrane voltage of these cells. However, whereas the mouse Slo3 channel is opened in response to a decrease in the concentration of protons within the sperm (i.e., an increase of the pH inside the cell), human Slo3 is largely controlled by changes in the levels of calcium ions. An increase in the calcium concentration within the cell opens the human Slo3 channel, more than a decrease in the proton concentration does.

Altogether, Brenker et al. identify Slo3 as the principal potassium channel in human sperm and reveal more fundamental differences between human sperm and mouse sperm. Thereby, this work further stresses the need to be cautious about using mice as a model of male fertility in humans.

sperm-specific Slo3 channel. Deletion of the Slo3 gene abolishes $I_{KSper}$ (*Santi et al., 2010*; *Zeng et al., 2011*, *2013*); male $Slo3^{-/-}$ mice are infertile due to defects in sperm motility (*Santi et al., 2010*; *Zeng et al., 2011*), osmoregulation (*Santi et al., 2010*; *Zeng et al., 2011*), and acrosomal exocytosis (*Santi et al., 2010*).

In humans, it is unknown whether Slo3 is functionally expressed in sperm and serves a similar key role for fertilization. Here, we examine the properties of human sperm $K^+$ current by patch-clamp recording and also define properties of currents arising from heterologous expression of hSlo3 and its auxiliary subunit hLRRC52 (*Yang et al., 2011*). We find that human $I_{KSper}$ and heterologously expressed human Slo3 currents share similar biophysical properties, pharmacology, and ligand dependence. Furthermore, we identify Slo3 and LRRC52 proteins in human sperm. Remarkably, whereas mouse Slo3 is exclusively controlled by $pH_i$ (*Schreiber et al., 1998*; *Zhang et al., 2006a*; *Yang et al., 2011*; *Zeng et al., 2011*), activation of human Slo3 is regulated by $[Ca^{2+}]_i$ and also, more weakly, by cytosolic alkalization. These results show that, between mouse and human sperm, signalling pathways controlling the principal $K^+$ channel and, thereby, $V_m$ are also distinctively different.

## Results

### Identification of $I_{KSper}$ in human sperm

We recorded currents from human sperm by the patch-clamp technique (*Lishko et al., 2013*). Depolarizing voltage steps from a holding potential of −80 mV evoked outwardly rectifying voltage-gated currents (*Figure 1A,B*). At $pH_i$ 7.3, current amplitudes at −100 mV and 100 mV were −7.5 ± 5 pA and 80 ± 15 pA, respectively (n = 5) (mean ± SD; n = number of experiments) (*Figure 1F*). Several controls established that the currents are carried by $K^+$ channels and not by $Cl^-$ channels or CatSper (*Zeng et al., 2013*): lowering the extracellular $K^+$ concentration ($[K^+]_o$) from 150 to 5 mM shifted the reversal potential ($V_{rev}$) from 9.2 ± 1.5 mV to −16.5 ± 10 mV (n = 5) (*Figure 1B,C*). At low $[K^+]_o$, a decrease of extracellular $[Cl^-]_o$ did not change $V_{rev}$ any further (*Figure 1C*, *Figure 1—figure supplement 1A,B*), showing that currents are not carried by $Cl^-$ channels. Replacing intracellular $K^+$ by $Cs^+$ almost completely abolished outward currents at $V_m \leq 100$ mV (*Figure 1F*, *Figure 1—figure supplement 1C,D*). However, at $V_m \geq 100$ mV, residual $Cs^+$ outward currents persisted. In mouse

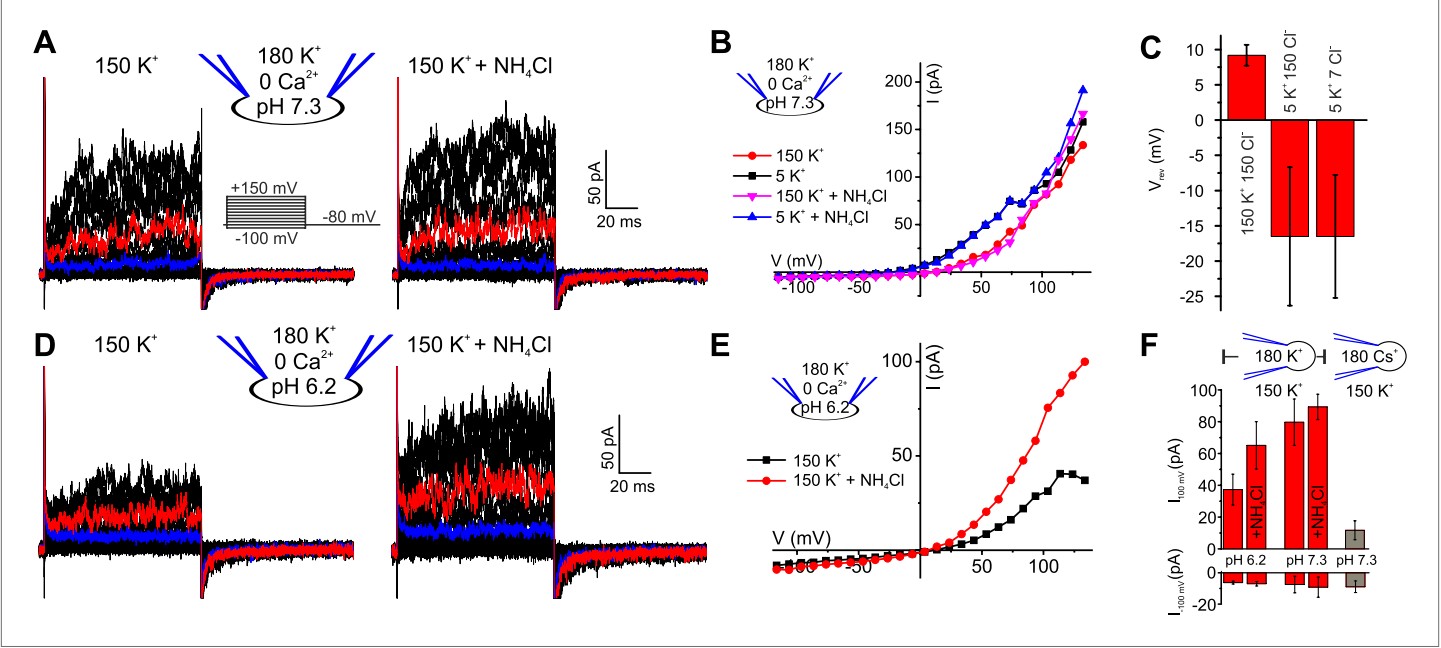

**Figure 1**. Voltage- and alkaline-activated K+ currents in human sperm. (**A**) Whole-cell currents before and after application of 10 mM NH4Cl. Traces at 35 mV and 85 mV are depicted in blue and red, respectively. (**B**) Current-voltage relation of recordings from (**A**) and currents recorded in 5 mM [K+]o. (**C**) Mean $V_{rev}$ of currents at $pH_i$ 7.3 in different extracellular solutions (n = 3–5). (**D**) Currents recorded at $pH_i$ 6.2. (**E**) Current-voltage relation of recordings from (**D**). (**F**) Mean currents before and after application of NH4Cl (10 mM) and with Cs+-based intracellular solution (180 mM Cs+) (n = 3–6).

The following figure supplements are available for figure 1:

**Figure supplement 1**. Voltage-gated currents in human sperm are carried by K+ channels.

Slo3−/− sperm, monovalent outward currents persisting at very positive $V_m$ are carried by CatSper (*Zeng et al., 2013*). Monovalent mouse and human CatSper current is suppressed by extracellular Ca2+ (*Kirichok et al., 2006*; *Lishko et al., 2011*; *Lishko et al., 2012*; *Zeng et al., 2013*). Consistent with CatSper channels conducting the residual Cs+ current in human sperm, current amplitudes at 120 mV were progressively suppressed by increasing extracellular Ca2+ (*Figure 1—figure supplement 1E,F*).

We conclude that monovalent cation currents at $V_m \leq 100$ mV are carried by K+-selective channels; we refer to this current as human $I_{KSper}$, analogous to the principal K+ current in mouse sperm. At more positive $V_m$, currents are also carried by CatSper channels.

## The principal K+ channel in human sperm is controlled by Ca2+ rather than $pH_i$

Mouse $I_{KSper}$ is strongly pH dependent (*Navarro et al., 2007*; *Santi et al., 2010*; *Zeng et al., 2011*; *Zhang et al., 2006a*). We, therefore, examined the pH sensitivity of $I_{KSper}$ and $V_m$ in human sperm. Decreasing pipette pH from 7.3 to 6.2 reduced outward currents by 2.2 ± 1.0-fold (n = 4) (*Figure 1A,D,F*). Moreover, intracellular alkalization by NH4Cl enhanced outward currents by 1.8 ± 0.9-fold (at 100 mV, n = 4) at pipette pH 6.2 (*Figure 1D–F*), but not at pipette pH 7.3 (*Figure 1A,B,F*). Thus, human $I_{KSper}$ is less sensitive to $pH_i$ than mouse $I_{KSper}$, which is enhanced about fourfold by increasing $pH_i$ from 6.5 to 7.5 (*Navarro et al., 2007*). In mouse sperm, $I_{KSper}$ sets $V_m$ in a strongly $pH_i$-dependent manner (*Navarro et al., 2007*; *Santi et al., 2010*; *Zeng et al., 2011*). Therefore, we tested under current-clamp whether, in human sperm, the control of $V_m$ by $I_{KSper}$ is dependent on $pH_i$. At pipette pH 6.2, $V_m$ was −23 ± 5 mV (n = 4) (*Figure 2A,C* left); raising pipette pH to 7.3 or alkalization by NH4Cl hyperpolarized sperm only slightly to −30 ± 5 mV and −31 ± 4 mV, respectively (n = 4) (*Figure 2A,B,C* left). At pipette pH 7.3, alkalization by NH4Cl did not further change $V_m$ (*Figure 2B,C* left). Of note, at $pH_i$ 6.2 and 7.3, $V_m$ was independent of [Cl−]o and dropped to about 0 mV at high [K+]o (*Figure 2A,B,C* left). In conclusion, under the recording conditions used here, $V_m$ of human sperm is only modestly $pH_i$ sensitive.

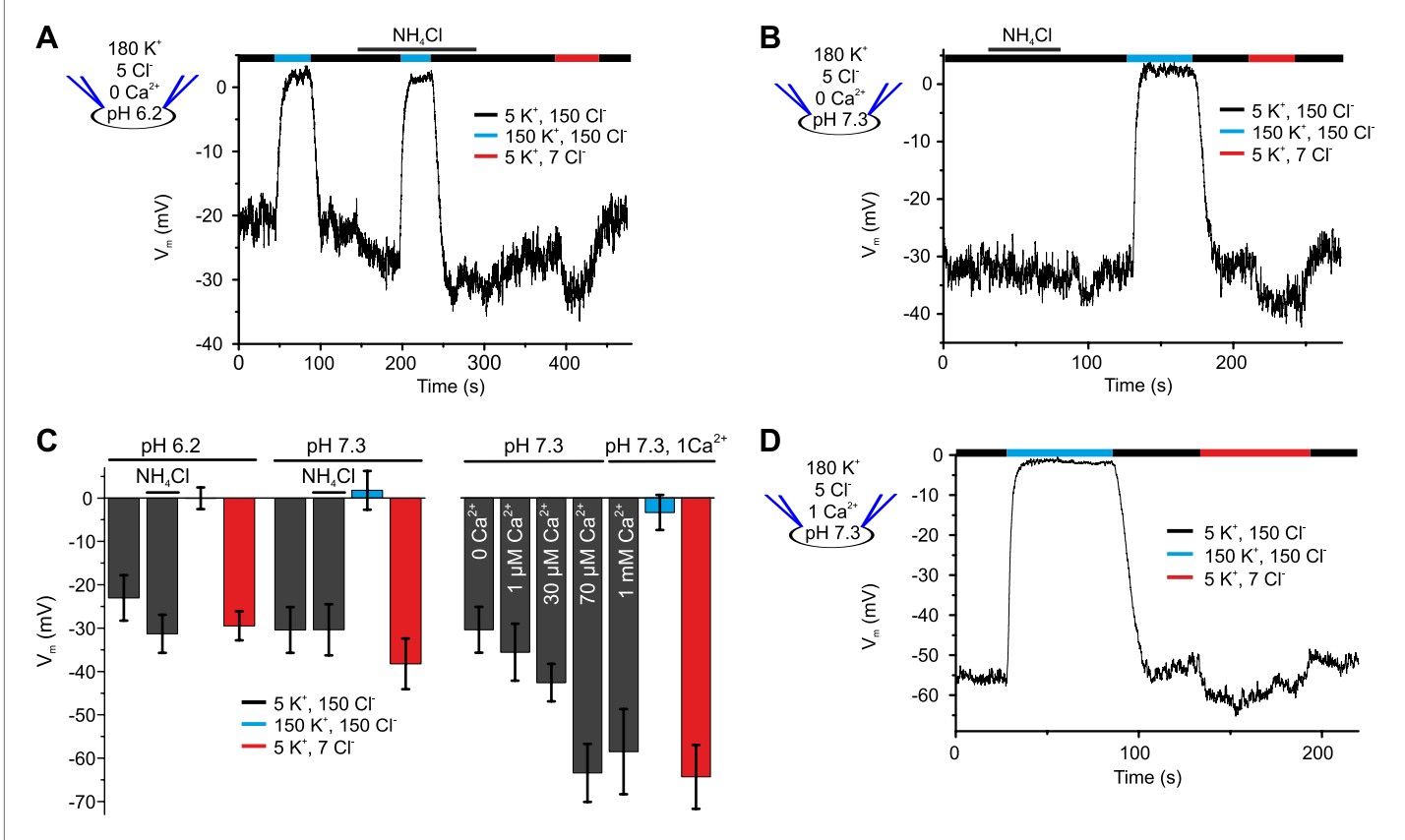

**Figure 2**. $V_m$ of human sperm is controlled by $[Ca^{2+}]_i$ rather than $pH_i$. (**A**) Current-clamp recording from human sperm ($pH_i$ 6.2) in extracellular solutions containing different $[K^+]$ and $[Cl^-]$ (in mM). Intracellular alkalization was evoked by superfusion with 10 mM $NH_4Cl$. (**B**) Current-clamp recording at $pH_i$ 7.3. (**C**) Left panel: mean $V_m$ under conditions as described in panel **A** and **B**; right panel: mean $V_m$ at indicated $[Ca^{2+}]_i$, and at 1 mM $[Ca^{2+}]_i$ under conditions described in panel **D** (n = 3–4). (**D**) Current-clamp recording at $pH_i$ 7.3 and 1 mM $[Ca^{2+}]_i$.

Given the modest effect of alkalization on human $K^+$ current, we tested whether $Ca^{2+}$ controls $I_{KSper}$ in human sperm. To compare current–voltage (I–V) relations at low and high $[Ca^{2+}]_i$ in the same cell, we studied voltage activation of $I_{KSper}$ before and after rapid photorelease of $Ca^{2+}$ from caged $Ca^{2+}$ (DMNP-EDTA) (**Figure 3A,B**), while monitoring $[Ca^{2+}]_i$ with the $Ca^{2+}$ indicator Fluo-4. Prior to $Ca^{2+}$ release, currents were similar to those recorded without $Ca^{2+}$ in the pipette solution (**Figure 3A,B**). Photorelease of $Ca^{2+}$ altered $I_{KSper}$ in several ways: currents activated more rapidly; at $V_m ≤ 70$ mV, amplitudes were enhanced and at $V_m ≥ 70$ mV, amplitudes saturated or even declined; thereby, the outward rectification was diminished; finally, $V_{rev}$ was shifted to more negative potentials (**Figure 3A,B**).

The $Ca^{2+}$ dependence of $I_{KSper}$ was quantified by recording the I-V relation at 2.5-1000 µM $Ca^{2+}$ in the pipette (**Figure 3C,D**). In the absence of $Ca^{2+}$, the mean current amplitude ($V_m = 3.5$ mV) was 3.5 ± 3 pA; increasing $Ca^{2+}$ to 40 µM and 1 mM $Ca^{2+}$ increased the amplitude to 14 ± 6 pA and 32 ± 17 pA, respectively (n = 3) (**Figure 3C**). For $[Ca^{2+}]_i > 10$ µM, current amplitudes were increased in a concentration-dependent manner, and the normalized I–V relation and $V_{rev}$ were shifted to more negative potentials (**Figure 3D**, **Figure 3—figure supplement 1C**). At $V_m ≳ 100$ mV, $Ca^{2+}$-activated currents levelled off or even declined. Control experiments showed that $Ca^{2+}$ activated $I_{KSper}$, and not $Ca^{2+}$-activated $Cl^-$ channels that were identified in human sperm (**Orta et al., 2012**): with 1 mM $[Ca^{2+}]$ in the pipette, a decrease of $[K^+]_o$ from 150 to 5 mM shifted $V_{rev}$ from −5.1 ± 10.3 mV to −52.9 ± 7.8 mV (n = 3) (**Figure 3E,F**); decreasing $[Cl^-]_o$ did not alter $V_{rev}$ (**Figure 3F**, **Figure 3—figure supplement 1A,B**).

Moreover, $Ca^{2+}$-activated $I_{KSper}$ was enhanced to some extent by alkalization. At $pH_i$ 6.2, $NH_4Cl$ increased the mean current amplitude from 52 ± 14 pA to 89 ± 8 pA (50 mV) and shifted $V_{rev}$ from −33 ± 7 mV to −54 ± 14 mV (n = 3) (**Figure 3G,H**). Thus, the enhancement of $I_{KSper}$ upon alkalization was similar in the presence and absence of intracellular $Ca^{2+}$.

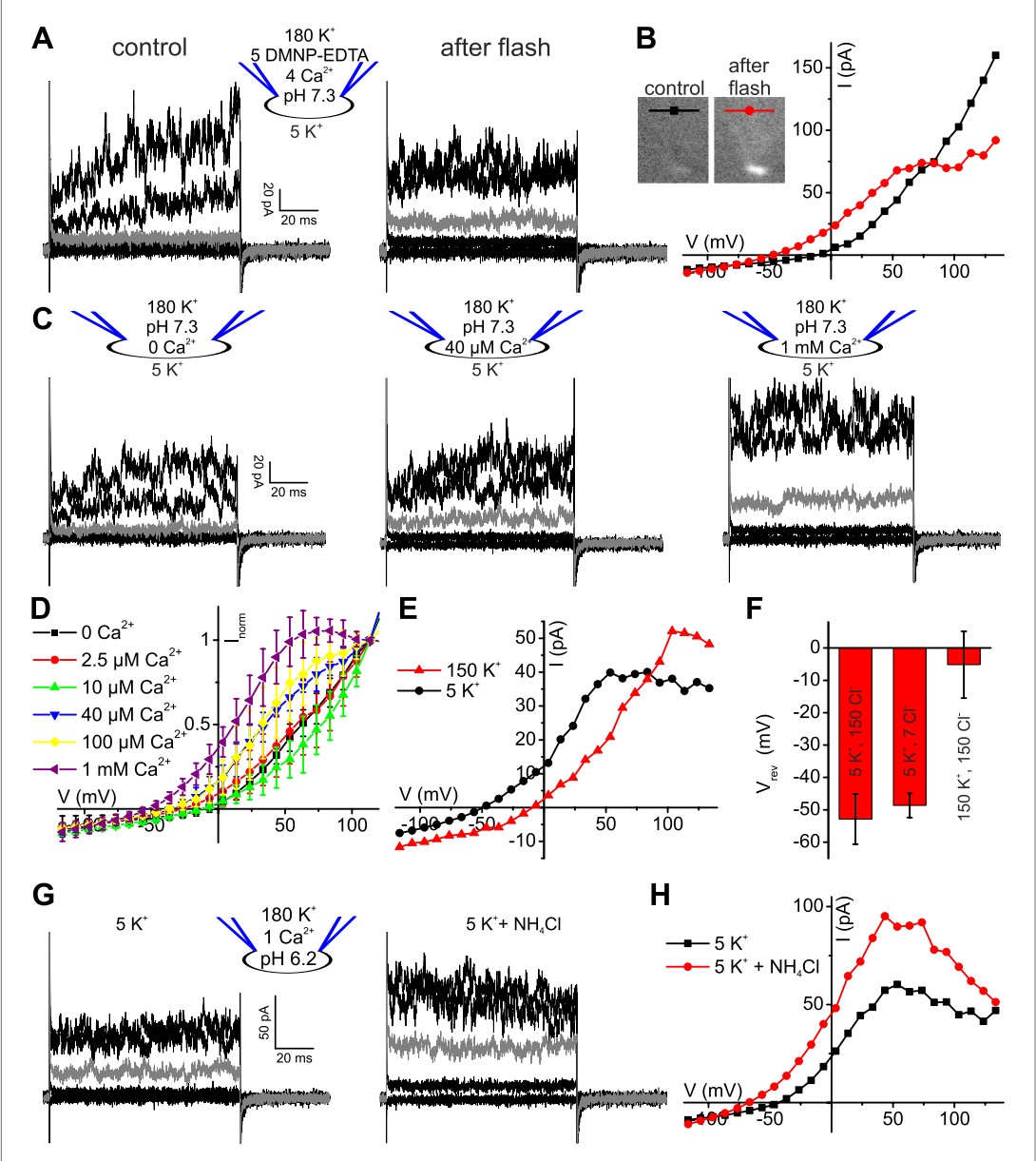

**Figure 3**. $Ca^{2+}$ enhances $K^+$ currents in human sperm. (**A**) Whole-cell currents recorded at $pH_i$ 7.3 with 5 mM DMNP-EDTA, 4 mM $Ca^{2+}$, and 10 µM Fluo-4 in 5 mM extracellular $K^+$ solution before (control) and after photorelease of $Ca^{2+}$; for simplification, only currents evoked by −105, −55, −5 (grey), 45, and 95 mV are depicted. (**B**) Current-voltage relation of recordings from **A**. (**C**) Whole-cell-currents at $pH_i$ 7.3 at 0 (left), 40 µM (middle) and 1 mM $[Ca^{2+}]_i$ (right) in extracellular solutions containing 5 mM $K^+$; for simplification, only currents evoked by −105, −55, −5 (grey), 45, and 95 mV are depicted. (**D**) Current-voltage relations of currents recorded at 2.5–1000 µM $[Ca^{2+}]_i$; I–V curves were normalized to the amplitude at 115 mV. (**E**) Current-voltage relation of currents recorded at $pH_i$ 7.3 and 1 mM $[Ca^{2+}]_i$; extracellular solutions contained 150 mM $K^+$ or 5 mM $K^+$. (**F**) Mean $V_{rev}$ of currents at $pH_i$ 7.3, 1 mM $[Ca^{2+}]_i$, and different $[K^+]_o$ and $[Cl^-]_o$ (in mM) (n = 3–4). (**G**) Whole-cell currents recorded at $pH_i$ 6.2, 1 mM $[Ca^{2+}]_i$, and 5 mM $[K^+]_o$, before and after superfusion with 10 mM $NH_4Cl$. (**H**) Current-voltage relation of recordings from (**G**).

The following figure supplements are available for figure 3:

**Figure supplement 1**. $Ca^{2+}$-activated currents in human sperm are carried by $K^+$ channels.

In current-clamp mode, $V_m$ was −30 ± 5 mV and −35 ± 7 mV at 0 and 1 µM $[Ca^{2+}]$ in the pipette, respectively; 30 µM and ≥ 70 µM $[Ca^{2+}]_i$ changed $V_m$ to −43 ± 4 mV and about −60 mV, respectively (n = 3-4) (*Figure 2C*, right). At high $[Ca^{2+}]_i$, $V_m$ was independent of $[Cl^-]_o$, but dropped to about 0 mV

at high $[K^+]_o$ (*Figure 2C right, D*). In conclusion, under the conditions used here, $V_m$ is set by $I_{KSper}$ that is controlled strongly by $Ca^{2+}$ and only modestly by $pH_i$.

## $Ca^{2+}$-activated $K^+$ currents exhibit hallmarks of Slo3 channels

The pH sensitivity of $I_{KSper}$, although modest, is reminiscent of the $I_{KSper}$ current in mouse sperm carried by Slo3 channels, whereas the $Ca^{2+}$ sensitivity is reminiscent of the prototypical $Ca^{2+}$-activated $K^+$ channel Slo1 (*Salkoff et al., 2006*). To identify the ion channel underlying human $I_{KSper}$, we tested several inhibitors of Slo3 and Slo1 channels at high $[Ca^{2+}]$ in the pipette, that is when $Ca^{2+}$-activated $K^+$ channels are strongly activated. The non-selective $K^+$ channel inhibitors quinidine and clofilium, previously shown to inhibit mouse $I_{KSper}$ (*Navarro et al., 2007*; *Zeng et al., 2011*) and heterologous Slo3 (*Tang et al., 2010*), abolished currents (*Figure 4A,B,E*). The inhibition by clofilium was irreversible (*Figure 4B*), which is a hallmark of its action on $I_{KSper}$ in mouse sperm (*Navarro et al., 2007*; *Zeng et al., 2011*). Moreover, perfusion of sperm with quinidine and clofilium depolarized the cell (*Figure 4F*). In contrast, tetraethylammonium (TEA) and iberiotoxin (IBTX), which block Slo1 but not Slo3 (*Tang et al., 2010*), neither affected $I_{KSper}$ nor $V_m$ of human sperm (*Figure 4C–F*). Although the Slo3 inhibitors employed are not selective for Slo3, the action of these drugs together with the ineffective Slo1 inhibitors provides critical evidence that $I_{KSper}$ is carried by Slo3, but not by Slo1.

Surprisingly, MDL12330A, an inhibitor of CatSper (*Brenker et al., 2012*), also blocked $Ca^{2+}$-activated $I_{KSper}$ and depolarized the cell under conditions (extra- and intracellular 1 mM $Ca^{2+}$) where CatSper currents are negligibly small (*Figure 4F,G*).

As a further signature of the $K^+$ channels, we compared single-channel currents of heterologous hSlo3 with those recorded in human sperm. At −60 mV and high intracellular $[Ca^{2+}]_i$, $K^+$ channels openings displayed a single-channel conductance of 60–70 pS (*Figure 4H–J*), that is similar to that of hSlo3 (70 pS; *Figure 6—figure supplement 1*; *Zhang et al., 2006b*), but not to that of Slo1 (280 pS) (*Dworetzky et al., 1994*). These results demonstrate that the channel underlying $I_{KSper}$ displays pharmacological and functional properties consistent with hSlo3.

## Human sperm express Slo3 and LRRC52

Next, we tested for the presence of Slo1 and Slo3 proteins in human sperm. Using targeted protein mass-spectrometry (MS) (*Figure 5A*, *Supplementary file 1*), we identified in purified human sperm 5 and 3 proteotypic peptides corresponding to Slo3 and its auxiliary subunit LRRC52, respectively. As positive controls, we identified the following proteins known to be expressed in human sperm: CatSper, the $Ca^{2+}$-ATPase PMCA4, the proton channel Hv1, $Na^+/K^+$-ATPase α4, and IZUMO. However, we did not detect Slo1. Similar results were obtained by shot-gun proteomics (*Wang et al., 2013*), which identifies Slo3 and other known components of human sperm, but not Slo1.

Moreover, we expressed hSlo3 heterologously with a hemagglutinin(HA)-tag in chinese hamster ovary (CHO) cells (*Figure 5—figure supplement 1*). In Western blots of hSlo3-transfected, but not of wild type cells, both anti-HA and anti-hSlo3 antibodies labelled polypeptides with an apparent molecular weight ($M_w$) of about 120 kDa (*Figure 5B*). The predicted $M_w$ of Slo3 is 130 kDa. In Western blots of human sperm, the anti-Slo3 antibody labelled polypeptides of ~125 kDa and ~80 kDa (*Figure 5B*); in both polypeptide bands, we confirmed by MS that the bands recognized by the antibody contained hSlo3. The 80 kDa polypeptide might be a product of the cleaved Slo3 channel. Post-translational cleavage has been reported also for other ion channels in ciliary structures (*Molday et al., 1991*; *Bönigk et al., 2009*). Finally, the anti-hSlo3 antibody stained the flagellum (*Figure 5C*), consistent with the localization of $I_{KSper}$ in mouse sperm (*Navarro et al., 2007*). Together, MS, Western blot-analysis, and immunocytochemistry show that human sperm express Slo3.

## Heterologous human Slo3 is activated by $Ca^{2+}$

Considering that mouse Slo3 is insensitive to $Ca^{2+}$ (*Schreiber et al., 1998*), the $Ca^{2+}$ activation of human $I_{KSper}$ is remarkable. Therefore, we studied human Slo3 co-expressed with hLRRC52 in *Xenopus* oocytes. First, we investigated the $pH_i$ sensitivity of hSlo3 in inside-out patches with step depolarizations between −60 and 260 mV from a holding potential of −140 mV. In the absence of $Ca^{2+}$, currents were only modestly activated at $pH_i$ 7 (*Figure 6A*, upper left), but enhanced at $pH_i$ 8 (*Figure 6A*, upper right). Similar to previous observations (*Leonetti et al., 2012*), an increase of $pH_i$ from 7 to either $pH_i$ 8 or 9 increased currents by 1.9 ± 0.4-fold and 2.2 ± 0.5-fold, respectively (200 mV; n = 4).

At $pH_i$ 7 and $pH_i$ 8, 60 μM $Ca^{2+}$ enhanced both outward currents and inward tail currents (*Figure 6A*, middle traces). At 300 μM $Ca^{2+}$, inward tail currents were further enhanced but outward currents were

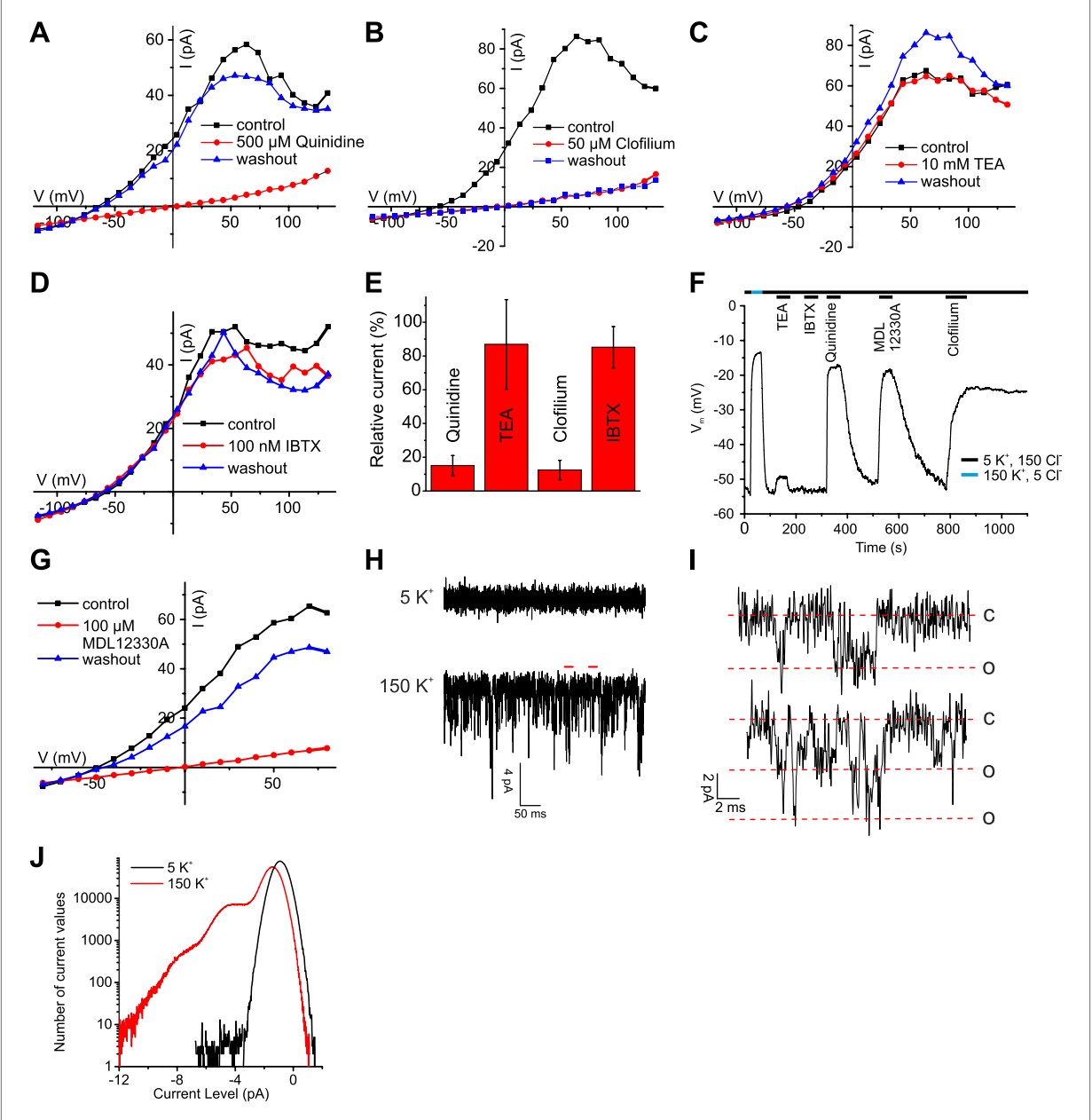

**Figure 4**. $Ca^{2+}$-activated $K^+$ currents in human sperm exhibit hallmarks of Slo3 channels. (**A–D**, **G**) Current-voltage relation of whole-cell currents from human sperm recorded in $K^+$-based intracellular solution at $pH_i$ 7.3 and 1 mM $[Ca^{2+}]_i$ in 5 mM extracellular $K^+$ before, during, and after application of inhibitors. (**E**) Mean outward currents at 65 mV in the presence of 500 μM quinidine, 10 mM TEA, 50 μM clofilium, or 100 nM IBTX. (**F**) Current-clamp recording from human sperm in intracellular solution ($pH_i$ 7.3) containing 180 mM $K^+$ and 1 mM $Ca^{2+}$. Sperm were bathed in extracellular solution containing 5 mM $K^+$ and 150 mM $Cl^-$. Concentrations of drugs were as in (**A–E**, **G**). (**H**) Current trace recorded at $pH_i$ 7.3 and 70 μM $[Ca^{2+}]_i$ at −60 mV in HS (top) and K-based HS (bottom) Filter: 2 kHz. (**I**) Segments indicated by the red bars in panel (**H**) shown on an extended time scale (5 kHz), revealing opening events of one (top) and two $K^+$ channels (bottom). Red lines correspond to conductance levels of 0 (c), 65 (o), and 130 pS (o). (**J**) Histogram of current amplitudes recorded in 5 mM $K^+$ and 150 mM $K^+$, at the conditions described in (**H**).

reduced at potentials ≥ 200 mV (**Figure 6A**, bottom traces), indicating a block of outward currents by $Ca^{2+}$. Conductance-voltage (G-V) curves generated from tail currents illustrate the activation of hSlo3 at $pH_i$ 7 and $pH_i$ 8 in the presence of 0, 60, and 300 μM $Ca^{2+}$ (**Figure 6B,C**). At $pH_i$ 7, raising $Ca^{2+}$ from 0 to 300 μM enhanced hSlo3 conductance by 6.6 ± 1.7-fold (200 mV), in contrast to the only 2.2-fold increase evoked by raising $pH_i$ from 7 to 9. Thus, $Ca^{2+}$ activates hSlo3 much more effectively than alkaline $pH_i$.

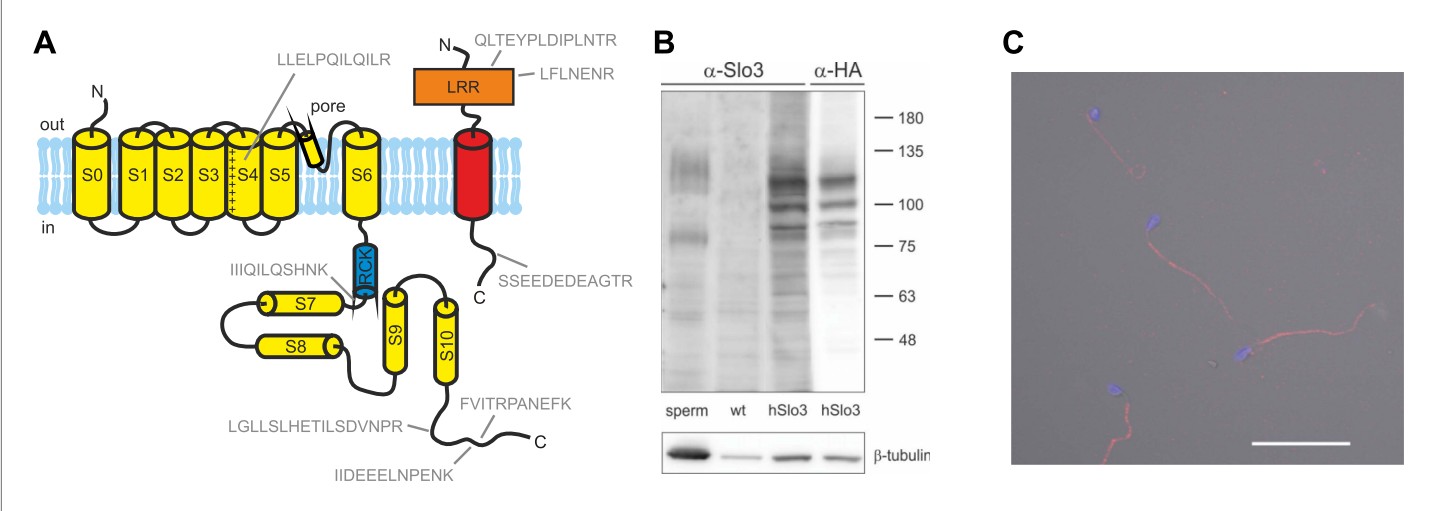

**Figure 5**. Human sperm express Slo3 and its auxiliary subunit LRRC52. (**A**) Predicted membrane topology of hSlo3 (yellow) and hLRRC52 (red) polypeptides. Proteotypic peptides identified in human sperm by targeted protein mass-spectrometry are indicated in grey. (**B**) Western blot of total proteins of human sperm, CHO cells (wt), and CHO cells transfected with HA-tagged hSlo3 (hSlo3). The Western blot was probed with an anti-hSlo3 and anti-HA antibody. The molecular masses (kDa) of the protein standard are indicated on the right. (**C**) Human sperm stained with an antibody directed against hSlo3 (red). The DNA in the head was stained with DAPI (blue). Scale bar: 30 µM.

The following figure supplements are available for figure 5:

**Figure supplement 1**. Specificity of anti-hSlo3 antibody.

---

Surprisingly, the amplitudes of $Ca^{2+}$-activated tail currents, whether at 60 or 300 µM $Ca^{2+}$, were similar at $pH_i$ 7 and $pH_i$ 8 (compare *Figures 6B and 6C*), suggesting that, at elevated $[Ca^{2+}]_i$, Slo3 is rather insensitive to changes in $pH_i > 7$. At $pH_i$ 8, we also examined the action of a broader range of $[Ca^{2+}]_i$. For $[Ca^{2+}]_i \geq 10$ µM, hSlo3 tail currents increased over at least three orders of magnitude of $[Ca^{2+}]$ (*Figure 6D*). Finally, Slo3 single-channel openings in patches held at −60 mV ($pH_i$ 8) were also markedly increased by 60 µM and 300 µM $Ca^{2+}$ (*Figure 6—figure supplement 1*), indicating that $Ca^{2+}$ enhances Slo3 activation also at physiological $V_m$.

Although the ranges of $pH_i$ and $Ca^{2+}$ concentrations that affect $I_{KSper}$ and hSlo3 expressed in oocytes are similar, compared to $I_{KSper}$, the I–V relation of hSlo3 currents appears shifted to more positive potentials (compare *Figure 1B,E and 3B* with *Figure 6B,C*). This difference might be due to the non-mammalian expression system, the excised-patch conditions, differences in ionic composition of solutions, or a combination of all of them. Therefore, we recorded whole-cell currents evoked by voltage steps in CHO cells that co-expressed hSlo3 and hLRRC52, using conditions similar to those used for sperm recordings. hSlo3 currents in CHO cells were also modestly enhanced by alkalization (*Figure 6—figure supplement 2*), and $Ca^{2+}$ shifted the I-V relation to more negative $V_m$ (*Figure 6E*, *Figure 6—figure supplement 2*). In the absence and presence of $Ca^{2+}$, the I–V relations of hSlo3 and $I_{KSper}$ were similar (*Figure 6F*).

Taken together, the heterologous expression demonstrates that human Slo3 is a $Ca^{2+}$-activated rather than a strictly alkaline-activated $K^+$ channel. These results strengthen our conclusion that Slo3 underlies the voltage-, $Ca^{2+}$-, and alkaline-activated $I_{Ksper}$ current in human sperm.

## Mouse Slo3 is not activated by intracellular $Ca^{2+}$

Although mouse Slo3 is insensitive to intracellular $Ca^{2+}$ (*Schreiber et al., 1998*), we wondered whether the auxiliary subunit LRRC52 (*Yang et al., 2011*) might confer $Ca^{2+}$ sensitivity on Slo3 channels. Therefore, we co-expressed mSlo3 with mLRRC52. Raising $pH_i$ from 7 to 8 strongly enhanced mSlo3 steady-state (*Figure 7A*) and tail (*Figure 7B*) currents, similar to previous results (*Schreiber et al., 1998*; *Zhang et al., 2006a*; *Yang et al., 2011*). At $pH_i$ 8, $Ca^{2+}$ (60 or 300 µM) did not enhance currents (*Figure 7A,B*). Instead, outward mSlo3 currents were strongly suppressed by $Ca^{2+}$ (*Figure 7A*) and even tail-current amplitudes were reduced (*Figure 7B*). A G-V plot derived from tail currents confirms that $Ca^{2+}$ does not activate mSlo3, but inhibits mSlo3 following activation by positive potentials

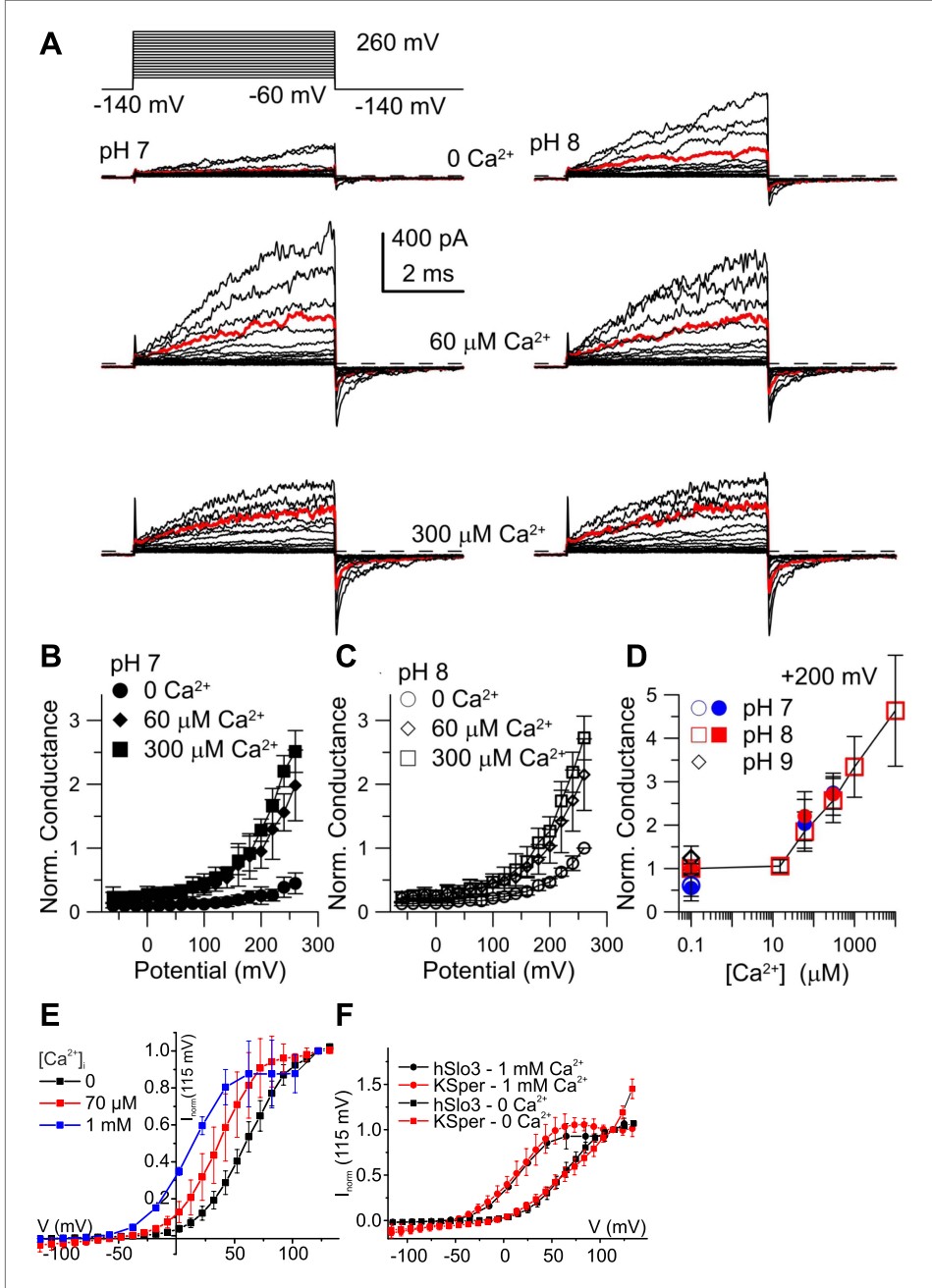

**Figure 6**. Activation of heterologous hSlo3 by intracellular $Ca^{2+}$. (**A**) Families of hSlo3 + hLRRC52 currents in oocytes at pH 7 and 8 with indicated $[Ca^{2+}]_i$. Current trace at +200 mV is depicted in red. (**B** and **C**) Current-voltage relations of tail currents determined at −140 mV; amplitudes were normalized to the amplitude evoked by step to 200 mV, 0 $[Ca^{2+}]_i$, and pH 8. (**D**) Tail current amplitudes (activated by 200 mV) as function of $[Ca^{2+}]_i$ for pH 7 and pH 8. Normalization as in (**B**). (**E**) Current-voltage relation of steady-state hSlo3 + hLRRC52 currents in CHO cells at $pH_i$ 7.3 and 0, 70 μM, and 1 mM $[Ca^{2+}]_i$. Currents were normalized to the amplitude evoked at 115 mV. (**F**) Current-voltage relation of hSlo3 + hLRRC52 currents in CHO cells and $I_{KSper}$ recorded from human sperm at 0 and 1 mM $[Ca^{2+}]_i$ ($pH_i$ 7.2). Normalization as in panel (**D**).

The following figure supplements are available for figure 6:

**Figure supplement 1**. $Ca^{2+}$ increases hSlo3 single-channel openings at −60 mV.

**Figure supplement 2**. Currents carried by hSlo3 co-expressed with hLRRC52 in CHO cells.

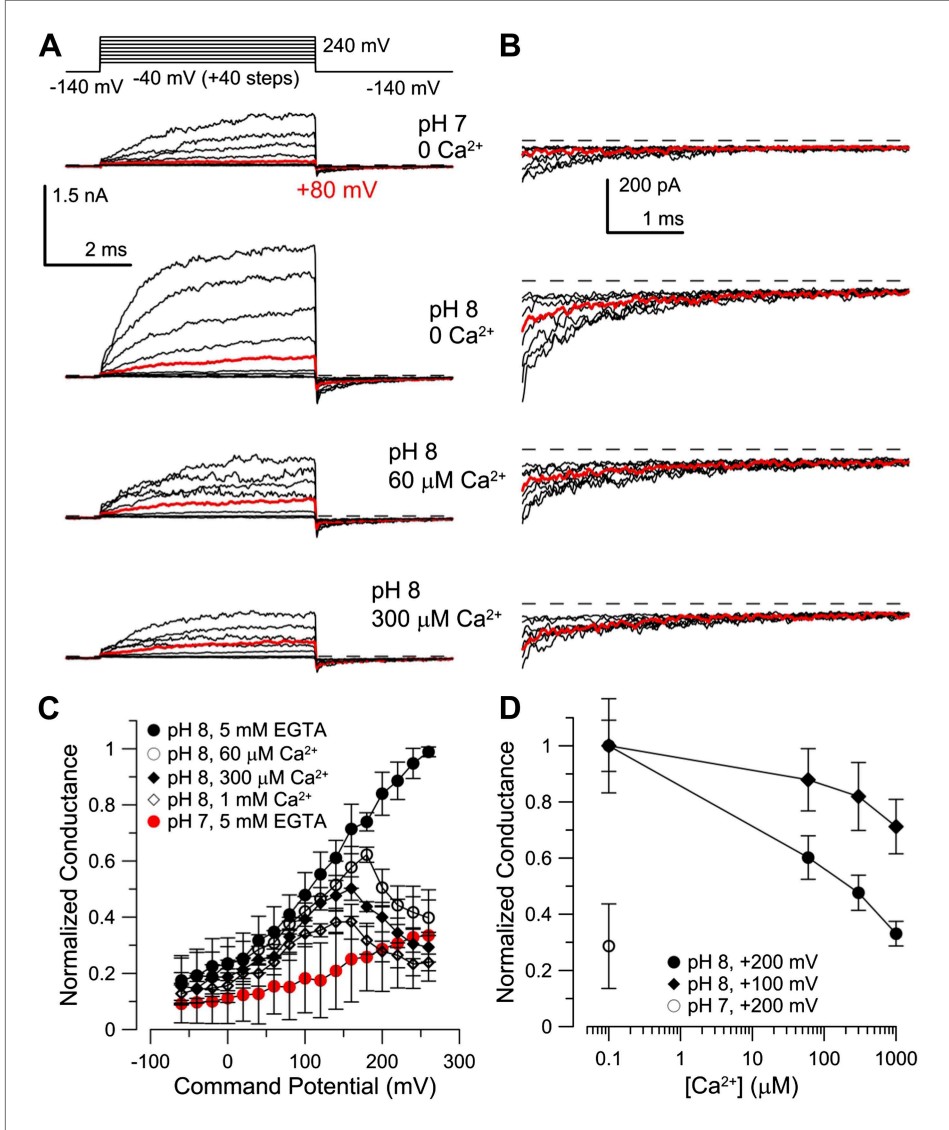

**Figure 7**. Co-expression of mLRRC52 does not confer $Ca^{2+}$ sensitivity on mSlo3. (**A**) Currents were activated with 0 $[Ca^{2+}]_i$ pH 7, 0 $[Ca^{2+}]_i$ pH 8, 60 µM $[Ca^{2+}]_i$ pH 8, and 300 µM $[Ca^{2+}]_i$ pH 8. Red trace corresponds to step to 60 mV. (**B**) Larger gain display of tail currents from (**A**). (**C**) Normalized current–voltage relations of tail currents determined at −140 mV; currents were normalized to tail current amplitude following the step to 200 mV. (**D**) Normalized tail-current amplitude as a function of $[Ca^{2+}]_i$. Tail currents were normalized to 0 $[Ca^{2+}]_i$ at pH 8.

The following figure supplements are available for figure 7:

**Figure supplement 1**. Co-expression of hLRRC52 with mSlo3 does not confer $Ca^{2+}$ sensitivity on mSlo3.

**Figure supplement 2**. mSlo3 is more sensitive to voltage-dependent block by $Ca^{2+}$ than hSlo3.

(*Figure 7C*). Currents carried by mSlo3 co-expressed with hLRRC52 were also not activated, but suppressed by $Ca^{2+}$ (*Figure 7—figure supplement 1*). These results exclude the possibility that hLRRC52 confers $Ca^{2+}$ sensitivity on hSlo3. Suppression of mSlo3 tail currents by $Ca^{2+}$ (*Figure 7C,D*) reflects persistent occupancy of the mSlo3 pore by $Ca^{2+}$ following repolarization. We note that the voltage-dependent suppression of currents by $Ca^{2+}$ is much more pronounced for mSlo3 compared to hSlo3. $Ca^{2+}$ occludes the pore of mSlo3 with about 10-fold higher affinity than that of hSlo3 (*Figure 7—figure supplement 2*); therefore, inhibition of hSlo3 by $Ca^{2+}$ occurs only at very positive, non-physiological $V_m$.

Together, our results show that regulation of mSlo3 and hSlo3 channels by cytosolic ligands is distinctively different, paralleling the differential regulation of $I_{Ksper}$ in mouse and human sperm by $pH_i$ and $Ca^{2+}$.

## Progesterone stimulates $Ca^{2+}$ levels sufficient to activate Slo3

In human sperm, the female sex hormone progesterone directly activates CatSper (*Lishko et al., 2011*; *Strünker et al., 2011*; *Brenker et al., 2012*; *Smith et al., 2013*). Progesterone-evoked $Ca^{2+}$ influx via CatSper has been implicated in sperm chemotaxis, hyperactivation, and acrosomal exocytosis (*Blackmore et al., 1990*; *Publicover et al., 2007*; *Publicover et al., 2008*). We examined whether stimulation of human sperm by progesterone enhances $Ca^{2+}$ levels sufficient to activate Slo3 channels. Sperm were loaded with $Ca^{2+}$ indicators of different $Ca^{2+}$ affinity as surrogates for high- to low-affinity $Ca^{2+}$-binding sites (*Figure 8A,C*). The progesterone-evoked transient $Ca^{2+}$ response was faithfully tracked by high- ($K_D = 0.35$ μM), moderate- ($K_D = 9.7$ μM), and low-affinity $Ca^{2+}$ indicators ($K_D = 90$ μM) (*Figure 8A*). The $Ca^{2+}$ ionophore ionomycin evoked a sustained fluorescence increase reflecting the indicator response to near saturating, millimolar $[Ca^{2+}]_i$ (*Figure 8A*). For indicators with a $K_D$ value $\leq 2.3$ μM, the amplitudes of progesterone- and ionomycin-induced $Ca^{2+}$ signals were similar (*Figure 8A,B*), suggesting that these high-affinity indicators become saturated with $Ca^{2+}$ during the progesterone-evoked response. For indicators with $K_D$ values $\geq 9.7$ μM, the amplitude ratio of progesterone-evoked/ionomycin-evoked $Ca^{2+}$ signals decreased with increasing $K_D$ values. However, even for the low-affinity indicator Fluo-5N, which reports $[Ca^{2+}]_i$ changes in a concentration range of about 9–900 μM, the amplitude

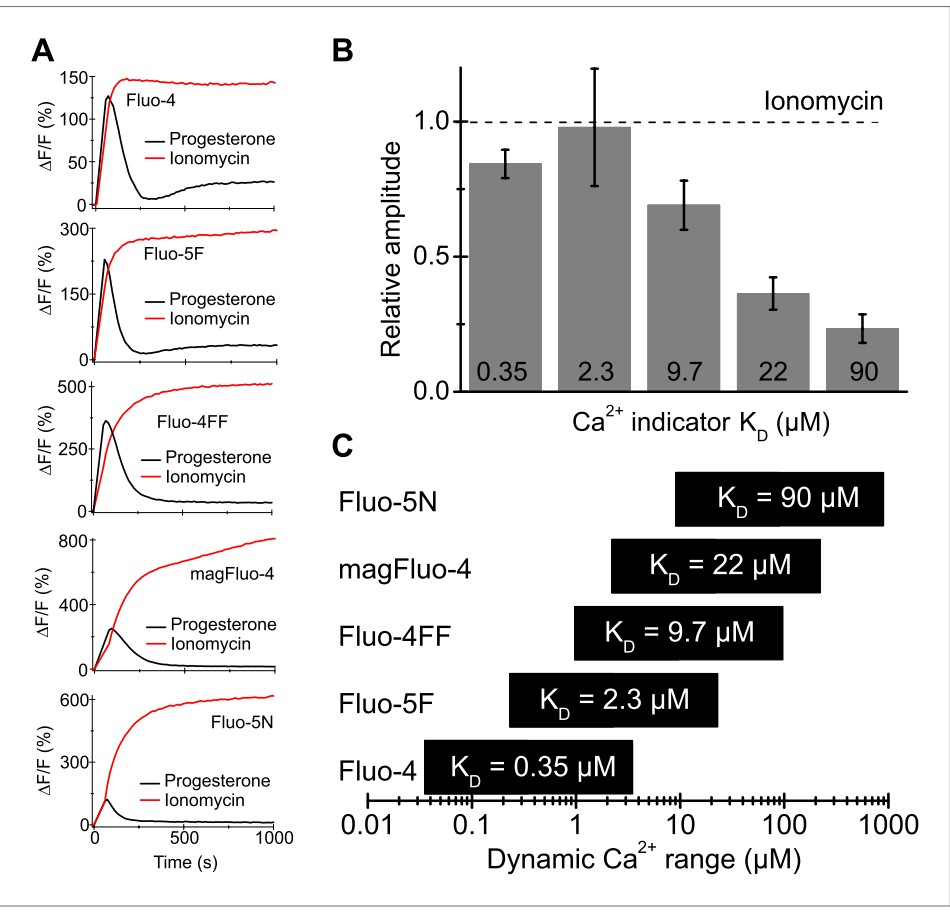

**Figure 8**. Progesterone-evoked $Ca^{2+}$ responses in human sperm. (**A**) $Ca^{2+}$ signal evoked by progesterone (2 μM) and ionomycin (2 μM) in sperm loaded with different $Ca^{2+}$ indicators. (**B**) Relative amplitude of the progesterone- vs ionomycin-induced $Ca^{2+}$ signal in sperm loaded with indicators of various $Ca^{2+}$ sensitivity. (**C**) Dynamic range of $Ca^{2+}$ sensitivity for different indicators assuming 1:1 binding of $Ca^{2+}$.

ratio was as large as 0.25. Thus, considering the dynamic range of indicators (*Figure 8C*), our results indicate that $Ca^{2+}$ levels reached during a physiological $Ca^{2+}$ response are sufficient to activate Slo3.

## Discussion

Here, we show that the biophysical and pharmacological properties of human $I_{KSper}$ conform with the properties of hSlo3, but not with those of hSlo1 or other members of the Slo family. First, hSlo3 and $I_{KSper}$ are modestly sensitive to $pH_i \leq 7.0$ and are more strongly activated by $Ca^{2+}$. Second, the I–V relations of hSlo3 and $I_{KSper}$ are similar, in the absence and presence of intracellular $Ca^{2+}$. Third, several Slo3, but not Slo1 inhibitors block $I_{KSper}$. Fourth, we identify the Slo3 protein and its auxiliary subunit LRRC52 in human sperm. Finally, both human $I_{KSper}$ (*Mannowetz et al., 2013*) and heterologously expressed hSlo3 (*Figure 9A,C*) are inhibited by progesterone; progesterone inhibits human $I_{KSper}$ and hSlo3 with constants of half-maximal inhibition ($K_i$) of 7.5 µM (*Mannowetz et al., 2013*) and 17.5 ± 2 µM (n = 3), respectively.

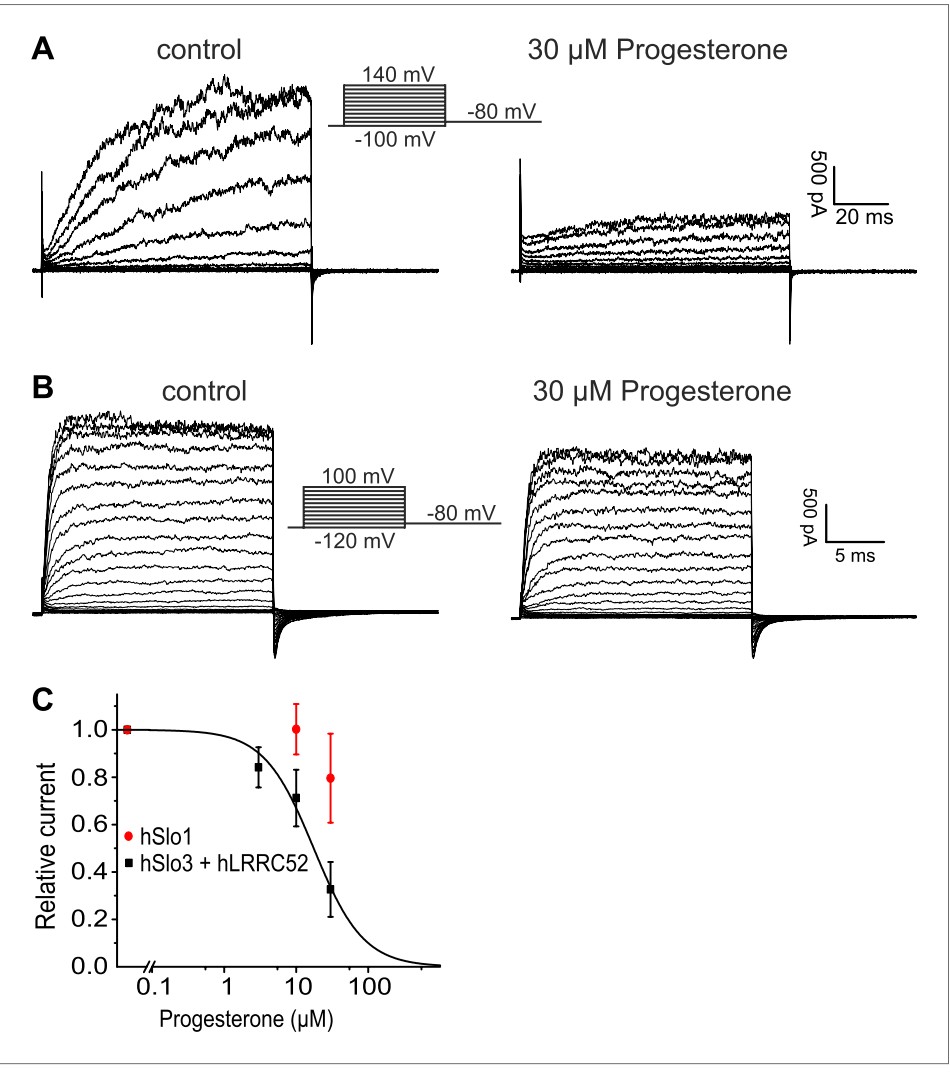

**Figure 9**. Progesterone inhibits hSlo3 but not hSlo1. (**A**) Whole-cell hSlo3 + hLRRC52 currents recorded in CHO cells at $pH_i$ 7.3 before and after perfusion with 30 µM progesterone. (**B**) hSlo1 currents recorded in outside-out patches excised from CHO cells at $pH_i$ 7.3 and 70 µM $[Ca^{2+}]_i$ before and after perfusion with 30 µM progesterone. (**C**) Relative amplitude of hSlo1 and hSlo3 + hLRRC52 currents at 80 mV in CHO cells in the presence of progesterone.

The following figure supplements are available for figure 9:

**Figure supplement 1**. Lack of homology among Slo3 sequences in the ligand-sensing cytosolic domain.

While this manuscript was under review, *Mannowetz et al. (2013)* reported that the prototypical $Ca^{2+}$-activated member of the Slo channel family, Slo1, is the principal $K^+$ channel in human sperm. Several observations strongly argue against this conclusion. First, Slo1 is inhibited rather than activated at alkaline pH (*Avdonin et al., 2003*). Second, specific inhibitors of Slo1 did not inhibit $I_{KSper}$. Third, human Slo1 is largely insensitive to progesterone (*Figure 9B,C*). Fourth, we and others (*Wang et al., 2013*) are unable to identify the Slo1 protein in human sperm. Fifth, if Slo1 carried the current of about 125–150 pA in human sperm (recorded at 100 mV in symmetrical $K^+$ and saturating $Ca^{2+}$) (*Figure 3A,C*), this would correspond to the opening of about 5–6 BK channels. Under such conditions, discrete opening and closing transitions of single BK channels would be readily visible; not only at 100 mV, but even more so at voltages < 100 mV. Thus, recordings of $I_{KSper}$ current in human sperm are consistent with lower conductance openings characteristic of Slo3, but not Slo1. Finally, the most obvious discrepancy between the two reports concerns the pharmacology. Mannowetz et al. show that $K^+$ currents in human sperm are abolished by the Slo1 inhibitors IBTX, charybdotoxin (CTX), and paxilline. Although we do not know the reason for this discrepancy, there are differences in experimental conditions. We tested the inhibitors at 1 mM extracellular $[Ca^{2+}]$ to prevent monovalent CatSper currents and at 1 mM intracellular $[Ca^{2+}]$ to strongly activate $I_{KSper}$. Mannowetz et al. tested the drugs at 100 µM extracellular $[Ca^{2+}]$ and in the absence of intracellular $Ca^{2+}$. Under these conditions, sizeable CatSper currents are recorded (*Figure 1—figure supplement 1E,F*), but activation of Slo1 would be minimal.

What might be the function of Slo3 in sperm? It has been suggested that the $I_{KSper}$-mediated hyperpolarization reinforces $Ca^{2+}$ influx via CatSper by increasing the electrical driving force (*Clapham, 2013*). Alternatively, we propose that the hyperpolarization serves as a negative feedback that decreases the open probability of CatSper and, thereby, curtails rather than enhances $Ca^{2+}$ influx. Why did Slo3 switch in human sperm from a strictly pH-sensitive to a pH- and $Ca^{2+}$-sensitive $K^+$ channel? In mouse, Slo3 and CatSper are both voltage- and strongly alkaline-activated. In human but not in mouse sperm, CatSper is directly activated by progesterone and prostaglandins (*Lishko et al., 2011*; *Strünker et al., 2011*; *Brenker et al., 2012*; *Smith et al., 2013*). Thus, human CatSper mediates a ligand- rather than an alkaline-activated $Ca^{2+}$ influx. Moreover, the pH sensitivity of both Slo3 and CatSper is considerably lower in human compared to mouse sperm, suggesting that $Ca^{2+}$ activation of Slo3 may have evolved in concert with ligand activation of CatSper. This co-evolution might ensure that ligand-evoked $Ca^{2+}$ influx via CatSper is coupled to the $Ca^{2+}$-controlled hyperpolarization via Slo3. Thus, by curtailing $Ca^{2+}$ influx via CatSper, $I_{KSper}$ may serve a similar role in both mouse and human sperm despite its differential regulation by intracellular ligands. The control of $V_m$ by $Ca^{2+}$ and $pH_i$ and the interplay of CatSper and Slo3 deserve further study in intact, freely moving human sperm with non-invasive and kinetic techniques, for example using voltage-sensitive dyes.

Our results suggest that during a progesterone response, global $Ca^{2+}$ levels can reach concentrations > 10 µM, sufficient to activate Slo3. CatSper and Slo3 are both located in the principal piece. Moreover, progesterone-induced $Ca^{2+}$ signals originate in the flagellum and propagate in a tail-to-head direction (*Servin-Vences et al., 2012*). Due to the miniscule flagellar volume (about 2.5 fl), opening a few CatSper channels, each conducting several thousand $Ca^{2+}$ ions per second, would increase local flagellar $[Ca^{2+}]$ to levels that should readily exceed global $[Ca^{2+}]_i$. The potential interplay of Slo3 and CatSper in sperm is reminiscent of the interplay in neurons between $Ca^{2+}$-activated $K^+$ channels and voltage-gated $Ca^{2+}$ channels ($Ca_v$) (*Prakriya and Lingle, 2000*). In neurons, Slo1 and $Ca_v$ channels interact to form local microdomains of $Ca^{2+}$ signalling near the plasma membrane (*Berkefeld et al., 2006*). In microdomains, $[Ca^{2+}]$ can rise to levels ≳ 100 µM that are readily sensed by $Ca^{2+}$-activated $K^+$ channels (*Rizzuto and Pozzan, 2006*). It needs to be shown whether Slo3 and CatSper are organized in similar microdomains.

Channels of the Slo family have been studied as models for allosteric regulation of gating by ligands (*Magleby, 2003*; *Lingle, 2007*). Slo channels and a large number of bacterial channels/transporters harbor a homologous octameric intracellular domain, dubbed the gating ring, that provides a template for ligand regulation of the pore domain (*Lingle, 2007*). The gating-ring motif has evolved for regulation of transmembrane ion flux by nucleotides, $Ca^{2+}$, $H^+$, $Na^+$, $Cl^-$, and probably other cytosolic ligands (*Salkoff et al., 2006*). Among different Slo isoforms, gating rings share structural similarities (*Wu et al., 2010*; *Leonetti et al., 2012*; *Yuan et al., 2012*). However, the conformational changes that couple binding to gating remain incompletely defined and the location of ligand-binding sites varies markedly (*Salkoff et al., 2006*). Our study adds an interesting twist, demonstrating that, even among Slo3 orthologues, the gating-ring motif has been exploited for regulation by different ligands — primarily $Ca^{2+}$ in

hSlo3 and primarily H[+] in mSlo3. Alignment of human Slo1 sequence with human Slo3 and other mammalian Slo3 sequences illustrates that, although the membrane-associated S0-S6 domain retains considerable identity (*Figure 9—figure supplement 1A*), there is extensive lack of identity in many segments of the ligand-sensing cytosolic domain (*Figure 9—figure supplement 1B*). Given that positions of ligand-sensing determinants in regulator of K[+] conductance (RCK)-containing proteins vary markedly within the RCK domain structures, it is not surprising that simple examination of the Slo3 sequence alignments does not reveal obvious determinants of either pH sensitivity in mSlo3 or $Ca^{2+}$-sensitivity in hSlo3.

At first glance, the fact that Slo3 orthologues differ in their ligand-dependence seems highly unusual. Yet, it is well-established that proteins essential for fertilization are rapidly evolving; orthologues often display a low degree of sequence similarity (*Swanson and Vacquier, 2002*; *Cai and Clapham, 2008*). For a set of mammalian species (human, bovine, mouse, dog, and opossum), the amino-acid identities (excluding a non-specific linker in the cytosolic domain) among Slo1 orthologues is typically > 92.6% (mSlo1 vs hSlo1: 99.5%). Slo3 orthologues exhibit identities in the range of 57–75% (mSlo3 vs hSlo3: 70.4%). Considering that the amino-acid identity between the pH-regulated mSlo3 and $Ca^{2+}$-regulated mSlo1 is 45.7%, the different ligand dependence between hSlo3 and mSlo3 is not so surprising.

## Materials and methods

### Materials and reagents

IBTX was purchased from Tocris (Minneapolis, MN, USA). DMNP-EDTA and $Ca^{2+}$ indicators were purchased from Invitrogen (Carlsbad, CA, USA). UPLC grade formic acid, acetonitrile and methanol were purchased from Biosolve (Valkenswaard, The Netherlands). All other reagents were obtained from Sigma-Aldrich (St. Louis, MO, USA).

### Sperm preparation

Samples of human semen were obtained from healthy volunteers with their prior consent. Sperm were prepared as described (*Strünker et al., 2011*). In brief, sperm were purified by a 'swim-up' procedure in human tubular fluid (HTF[+]) containing (in mM): 97.8 NaCl, 4.69 KCl, 0.2 $MgSO_4$, 0.37 $KH_2PO_4$, 2.04 $CaCl_2$, 0.33 Na-pyruvate, 21.4 lactic acid, 2.78 glucose, 21 HEPES, and 25 $NaHCO_3$ adjusted between pH 7.3-7.4 with NaOH. After washing, human serum albumin (HSA, 3 mg/ml; Irvine Scientfic, USA) was added to HTF[+] (referring to as HTF[++]). Sperm were incubated for at least 2 hr in HTF[++] at 37°C in a 10% $CO_2$ atmosphere. Under these conditions, sperm undergo capacitation. All recordings were done on capacitated sperm.

### Mass spectrometry

After purification by a 'swim-up' procedure, human sperm were lysed by several 'freeze/thaw' cycles and sonification steps in buffer containing (in mM): 10 HEPES pH 7.5, 2 EGTA, 1 DTT, protease inhibitor cocktails (Roche Applied Science and Sigma, Mannheim, Germany), and DNaseI (AppliChem, Darmstadt, Germany). Membranes were sedimented by centrifugation (100,000×*g*, 30 min, 4°C) and membrane proteins were processed by in-gel digestion, in-solution digestion, or a FASP protocol. Peptides were subjected to 1D-ESI-LC-MS/MS using a nanoAcquity UltraPerformance LC System (Waters, Milford, Massachusetts, USA) coupled to an LTQ Orbitrap Velos or Elite instrument (Thermo, Waltham, Massachusetts, USA). The resulting tandem MS data were searched using the Sequest algorithms embedded in Proteome Discoverer 1.2 (Thermo) against a SwissProt/UniProtKB human protein sequence database (including 56,582 entries). The mass tolerance for precursor ions was set to ≤10 ppm; the mass tolerance for fragment ions was set to ≤ 1 amu. For search result filtering, the false discovery rate (FDR) was set to < 1% and only peptides with search result rank 1 were accepted for identification. For targeted mass spectrometry, the instrument control software used a list of theoretical tryptic peptide masses for the proteins of interest for subsequent CID fragmentation, that is once a mass from the list was detected in the orbitrap full scan, it was preferred over all other co-eluting masses, independent of its signal intensity.

### Antibodies

Primary antibodies: Slo3 (Abcam, Cambridge, UK, catalog no. ab104630; for ICC, 1:50,000, for WB, 1:100), HA (Roche Applied Science, catalog no. 11867431001; for ICC, 1:1,000, for WB, 1:5000). Secondary antibodies: for ICC, rabbit Cy3-conjugated and rat Cy5-conjugated antibodies (1:400; Dianova, Hamburg, Germany); for WB, rabbit-HRP and rat-HRP (1:5000; Dianova).

## Preparation of recombinant constructs

EST clone BC028701 was obtained from Open Biosystems and verified by sequencing. Based on analysis of the NCBI *slo3* gene (*kcnu1*, NM_001031836), 62 base pairs (bp) present around the S10 region in BC028701 correspond to an intron left over from incomplete mRNA splicing. The intron sequences were removed by site-directed mutagenesis. In addition, there are at least three polymorphic sites present in the EST clone corresponding to amino-acid positions 192, 739, and 768 of hSlo3. The three sites were changed to match those of the human genomic sequence (NCBI reference sequence: NC_000008.10; Chromosome 8). This results in 192W, 739R, and 768W. The full length coding sequence of hSlo3 was subcloned into the oocyte expression vector pXMX (see details in *Tang et al., 2010*). The hSlo3 sequence used here is identical to that used in another study (*Leonetti et al., 2012*). An hLRRC52 (NM_001005214.3) clone was generated from two HEK genomic DNA fragments of 622 bp and 320 bp which correspond to hLRRC52 exons 1 and 2, respectively. The two fragments were amplified via over-lapping PCR and subcloned into the pXMX vector. For the mCherry-tagged hLRRC52, seven glycines were added as a spacer between the carboxy-terminus of hLRRC52 and the amino-terminus of mCherry; the hLRRC52-mCherry construct was also subcloned into pXMX vector. All cDNA clones were verified by sequencing. For expression in oocytes, cRNA was synthesized by SP6 polymerase after the cDNA template was linearized with the restriction enzyme MluI.

For expression in CHO cells, the full length coding sequence of hSlo3 was amplified from human testis cDNA. A perfect Kozak consensus sequence preceding the start codon and a sequence coding for a carboxy-terminal hemagglutinin tag (HA-tag) were added. The coding sequence harbored an arginine at the polymorphic site at position 768. The DNA was subcloned into a pcDNA3.1(+) vector (Invitrogen); the sequence coding for the neomycin resistance gene was replaced by the coding sequence for either citrine or EGFP. The hLRRC52-mCherry construct described above was also subcloned into the pcDNA3.1(+) vector for expression in CHO cells. All cDNA clones were verified by sequencing.

## Immunocytochemistry

Human sperm and CHO cells expressing hSlo3 were immobilized on glass coverslips and fixed for 3 min with 4% paraformaldehyde in phosphate buffered saline (PBS) containing (in mM): 137 NaCl, 2.7 KCl, 6.5 $Na_2HPO_4$, 1.5 $KH_2PO_4$, pH 7.4. To block nonspecific binding sites, cells were incubated for 20 min with blocking buffer (0.5% Triton X-100 and 5% chemiblocker (Merck Millipore, Germany) in PBS). Primary antibodies were diluted in blocking buffer and cells were incubated for 1 hr at room temperature. After washing with PBS, cells were incubated with fluorescent secondary antibodies in blocking buffer containing 0.5 µg/µl 4′,6-diamidino-2-phenylindole (DAPI; Invitrogen). After washing with PBS, cells were mounted on slides and examined with a confocal microscope (FV1000; Olympus, Düsseldorf, Germany).

## Protein preparation and western blot analysis

Wild type CHO cells and cells expressing the human Slo3 channel were lysed in a buffer containing 10 mM Tris/HCl, pH 7.5, 140 mM NaCl, 1 mM EDTA, 1% Igepal CA-630, 0.5% sodium deoxycholate, 0.1% SDS, and mammalian protease inhibitor cocktail (mPIC, Sigma) and incubated on ice for 30 min. The suspension (total lysate) was centrifuged for 5 min at 10,000×*g* (4°C) and the supernatant was used for WB analysis. Human sperm ($5 \times 10^6$) were resuspended in 2x SDS sample buffer containing β-mercaptoethanol. All samples were heated for 5 min at 95°C and separated by 4–12% SDS–polyacrylamide gel electrophoresis. For WB analysis, proteins were transferred onto PVDF membranes, probed with antibodies, and analysed using the LAS-3000 System (Fujifilm).

## Patch-clamp recording

We recorded from sperm in the whole-cell configuration as described (*Strünker et al., 2011*). Seals between pipette and sperm were formed either at the cytoplasmic droplet or the neck region in standard extracellular solution (HS) containing (in mM): 135 NaCl, 5 KCl, 1 $MgSO_4$, 2 $CaCl_2$, 5 glucose, 1 Na-pyruvate, 10 lactic acid, and 20 HEPES adjusted to pH 7.4 with NaOH. K-based HS contained (in mM): 135 KCl, 5 NaCl, 1 $MgSO_4$, 2 $CaCl_2$, 5 glucose, 1 Na-pyruvate, 10 lactic acid, and 20 HEPES adjusted to pH 7.4 with KOH. HS with low [Cl⁻] contained (in mM): 135 Na-aspartate, 5 KCl, 1 $MgSO_4$, 2 $CaCl_2$, 5 glucose, 1 Na-pyruvate, 10 lactic acid, and 20 HEPES adjusted to pH 7.4 with NaOH. The following pipette (10–15 MΩ) solutions were used (in mM): Cs-based IS: 130 Cs-asparate, 5 EGTA, 5 CsCl, and 50 HEPES at pH 7.3 with CsOH; K-based IS 0 $Ca^{2+}$: 130 K-aspartate, 10 NaCl, 1 EGTA,

and 50 HEPES at pH 7.3 with KOH; K-based IS with 100 μM or 1 mM $Ca^{2+}$: 130 K-aspartate, 7 NaCl, 2 KCl, 100 μM or 1 mM $CaCl_2$, and 50 HEPES at pH 7.3 with KOH; K-based IS with 40 μM $Ca^{2+}$: 140 K-aspartate, 50 HEPES, 10 NaOH, 5 KCl, 3 NTA, 1.3 $CaCl_2$, pH 7.3 with KOH; K-based IS with 10 μM $Ca^{2+}$: 130 K-asparate, 10 NaOH, 1 KCl, 3 HEDTA, 2 $CaCl_2$, and 50 HEPES at pH 7.3 with KOH. All voltages were corrected for the liquid junction potential of 16.5 mV. The osmolarity of intracellular and extracellular solutions was ~120 mOsm.

For functional studies in *Xenopus* oocytes, RNAs for hSlo3 and hLRRC52 were injected at a ratio of 1:1 by weight. Gigaohm seals were formed while oocytes were bathed in frog Ringer (in mM): 115 NaCl, 2.5 KCl, 1.8 $CaCl_2$, and 10 HEPES at pH 7.4. Pipette resistance was typically 0.8–2 MΩ after fire-polishing and filling with pipette solution containing (in mM): 140 K-methanesulfonate (MES), 20 KOH, 2 $MgCl_2$, and 10 HEPES at pH 7. The standard solution bathing excised inside-out patches was (in mM): 140 K-MES, 20 KOH, 5 mM EGTA, and 10 HEPES with pH adjusted as indicated. For addition of $Ca^{2+}$ to bathing solutions, in most cases free $[Ca^{2+}]$ was sufficiently high that buffering was not required (60 μM, 300 μM, 1 mM, and 10 mM). In one test solution, we omitted EGTA so that the estimated free $[Ca^{2+}]$ was defined presumably by the level of contaminant $Ca^{2+}$ in our water (estimated to be 10–15 μM based on calibration with a $Ca^{2+}$ electrode). Osmolarity for solutions was approximately 310 mOsm.

Solutions were applied directly to patches via an SF-77B fast perfusion stepper system (Warner Instruments, Hamden, CT, USA). $K^+$ currents were recorded from inside-out patches with an Axopatch 200B amplifier (Molecular Devices, Sunnyvale, CA, USA) and low-pass filtered at 10 kHz with an integral four-pole Bessel filter. Signals were digitized with a Digidata 1322A data acquisition system (Molecular Devices) at 100 kHz. Recordings were controlled by the pClamp 9.2 software suite (Molecular Devices). Capacitative component of currents was subtracted using the appropriately scaled trace for a step from −100 mV to 0 mV. All experiments were performed at room temperature (21–24°C).

CHO cells were co-transfected with hSlo3 and hLRRC52-mCherry constructs. Seals between pipette and cell were performed in standard extracellular solution (ES) containing (in mM): 140 NaCl, 5.4 KCl, 1 $MgCl_2$, 1.8 $CaCl_2$, 10 D-glucose, and 5 HEPES adjusted to pH 7.4 with NaOH. For the recordings in high $K^+$ solution, NaCl was exchanged by KCl and pH was adjusted to 7.4 with KOH. To study activation of Slo3 by alkalization, the following pipette (4–5 MΩ) solution was used (in mM): 130 K-aspartate, 10 NaCl, 1 EGTA, 5 HEPES, 15 D-glucose, pH was adjusted either to 6.2 or 7.3 using KOH. To study the activation of Slo3 by $Ca^{2+}$, the following pipette (2–3 MΩ) solutions were used (in mM): divalent-free solution: 130 K-aspartate, 10 NaCl, 1 EGTA, and 20 HEPES adjusted to 7.3 using KOH; 70 μM $Ca^{2+}$ intracellular solution: 130 K-aspartate, 10 NaCl, 0.5 $CaCl_2$, 1 NTA, and 20 HEPES adjusted to pH 7.3 using KOH, the final $Ca^{2+}$concentration was confirmed using the $Ca^{2+}$ dye Mag-Fura 2; 1 mM $Ca^{2+}$ intracellular solution: 130 K-aspartate, 10 NaCl, 1 $CaCl_2$, and 20 HEPES adjusted to pH 7.3 using KOH. Series resistance and cell capacitance were compensated to 70–85%. Voltages were corrected for the liquid junction potential. All recordings were performed at 20–22°C.

## Measurement of changes in $[Ca^{2+}]_i$

Changes in $[Ca^{2+}]_i$ were measured in 384 multi-well plates in a fluorescence plate reader (Fluostar Omega, BMG Labtech, Ortenberg, Germany) at 30°C. Sperm were loaded with the respective fluorescent $Ca^{2+}$ indicator (10 μM) (Molecular Probes) in the presence of Pluronic F127 (0.1% wt/vol) for 45 min at 37°C. After incubation, excess dye was removed by a centrifugation step (700×*g*, 10 min, RT). The sperm pellet was resuspended in HTF++ and equilibrated for 5 min at 37°C. Each well was filled with 50 μl of the sperm suspension (1 × $10^7$ sperm/ml); the fluorescence was excited at 480 nm and emission was recorded at 520 nm with bottom optics. Fluorescence was recorded before and after injection of 10 μl (1:6 dilution) of progesterone or ionomycin in HTF++. The solutions were injected into the wells manually with an electronic multichannel pipette.

## Data analysis

Statistical analysis and fitting of the data were performed using OriginPro 8.1 G SR3 (OriginLab Corporation, USA) or Clampfit 10.2 (Molecular Devices). All data are given as mean ± standard deviation (number of experiments).

## Acknowledgements

This work was supported by the German Research Foundation (SFB645) and National Institutes of Health (GM066215 and GM081748 to CJL). We thank Heike Krause for preparing the manuscript. We

thank Yuriy Kirichok and Polina Lishko for introducing Christoph Brenker into patch-clamp recordings from sperm.

## Additional information

### Funding

| Funder | Grant reference number | Author |
|---|---|---|
| German Research Foundation | SFB645 | Astrid Müller, U Benjamin Kaupp, Timo Strünker |
| National Institutes of Health | GM066215; GM081748 | Christopher J Lingle |

The funders had no role in study design, data collection and interpretation, or the decision to submit the work for publication.

### Author contributions

CB, TS, Conception and design, Acquisition of data, Analysis and interpretation of data, Drafting or revising the article; ZY, AM, FAE, CT, AP, WB, CJL, Acquisition of data, Analysis and interpretation of data, Drafting or revising the article; XX-M, Drafting or revising the article, Contributed unpublished essential data or reagents; UBK, Analysis and interpretation of data, Drafting or revising the article

## Additional files

### Supplementary files

• Supplementary file 1. Indicators of merit for the mass spectrometric results.

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
