## [Decision Letter]

[Editors’ note: a previous version of this study was rejected after peer review, but the authors submitted for reconsideration. The two decision letters after peer review are shown below.]

Thank you for choosing to send your work entitled “Slo3 in human sperm – a K^+^ channel activated by Ca^2+^” for consideration at *eLife*. Your full submission has been evaluated by a Senior editor and 3 peer reviewers, one of whom is a member of our Board of Reviewing Editors, and the decision was reached after discussions between the reviewers. We regret to inform you that your work will not be considered further for publication.

As can be seen from the enclosed reviews, there are substantial concerns about the data and conclusions, primarily concerning the sperm patch clamp data. The heterologous expression data showing calcium activation of hSlo3 are, on the other hand, convincing and quite interesting.

*Reviewer 1*:

The authors present compelling evidence that that Ksper current in human sperm is carried by the Slo3 potassium channel, as in mouse sperm, but the channel is gated by calcium, unlike in mouse where it is gated by protons. Heterologous expression studies confirm that human Slo3 seems to be calcium gated.

The results suggest that the Slo3 channel is gated by different ligands in different species. It is puzzling that the authors agree with the results of the Lishko lab that human Ksper is activated by calcium, but disagree as to which channel it is, as the Lishko paper argues that the mouse Ksper is in fact the (normally calcium gated) Slo1. I find no reason to doubt either study's conclusions and I expect the controversy to be settled with future work.

I believe the paper to be appropriate for publication in *eLife*, especially as the Lishko paper appeared there also.

*Reviewer 2*:

In the present manuscript, Struenker and colleagues describe functional differences between mouse and human Slo3 K^+^ channels expressed in spermatozoa. The authors provide experimental evidence for Slo3 expression in human spermatozoa and emphasize relevant functional differences: In contrast to mouse, hSlo3 is more sensitive to calcium than to intracellular alkalinisation. Functional hallmarks of hSlo3 as seen in sperm can be recapitulated in CHO cells expressing recombinant hSlo3 and the auxiliary subunit LRRC52. The authors put forward the proposal that during evolution the increased calcium sensitivity of hSlo2 has evolved in concert with the ligand sensitivity of human CatSper.

In principle, this is a carefully performed study that contributes to our understanding to the regulation of ion fluxes in human sperm. The major tenet of this manuscript, i.e., distinct and relevant species differences between human and mouse sperm, is about to impact the use of mouse sperm as a model system for humans.

However, mechanistically and conceptually the advances of the current manuscript are limited. The salient role of Slo3 in mouse sperm has recently been highlighted by several groups. It is well documented in the literature that mouse Slo3 is rather insensitive to calcium and that human Slo3 coexpressed with LRRC52 is less sensitive to alkaline pH as compared to mouse. The authors confirm these observation by others by direct electrophysiological recordings in human sperm. However, no information is provided on the molecular basis underlying these functional differences. Thus, the present manuscript still leaves central mechanistic questions unanswered and the paper is confirmatory to quite some extent.

*Reviewer 3*:

There are three major conclusions in this paper. First, the human K^+^ current I_Ksper_ is activated by intracellular Ca^2+^, rather than pH increase as previously established in mice. Second, Ca^2+^ increase in human sperm under “physiological” conditions (presumably through Catsper) is sufficient to activate I_Ksper_. Third, heterologously expressed human Slo3 (hSlo3) is activated by high [Ca^2+^]_i_, but is less sensitive to pH. These findings, if confirmed, represent a major step forward in our understanding of sperm membrane potential regulation and fertilization. The third conclusion is very well supported by careful and well-controlled analysis using heterologous expression in both Xenopus oocytes and CHO cells, with or without the auxiliary subunit LRRC52. I find that the first two conclusions, though quite interesting, are not convincing at their present form.

The major data supporting the conclusion that human I_Ksper_ is potentiated by Ca^2+^ but not pH_i_ is in Figures 2 and 3. In Figure 2 where the authors want to show what Ca^2+^, not pH, controls V_m_ in human sperm, two [Ca^2+^]_i_ conditions are compared: 0 Ca^2+^ and 1 mM Ca^2+^. The 0 Ca^2+^ pipette had 1 mM EGTA with no added Ca^2+^, which results a free [Ca^2+^] of perhaps < 10 nM. The pipette solution contains no ATP or GTP. Such condition may be fine for studying the biophysical properties of a channel with voltage-clamping, but is too “unphysiological” for studying the regulation of membrane potential. It is hard to interpret the apparent lack of large effect on V_m_ by pH_i_ change under this condition. The authors also cited the paper by Navarro (PNAS 104:7688-7692) to compare the difference between human and mouse sperm in the effects of pH_i_ on V_m_. However, there is large difference between the pipette solutions used in the two studies (Navarro et al. included ATP and GTP in the pipette and had apparently different free Ca^2+^ concentration). The authors should at least test several Ca^2+^ concentrations close to the physiological levels. In addition, all the pipette solutions have 50 mM HEPES. It's not clear whether 10 mM NH_4_Cl in the bath is able to significantly increase pH_i_ when 50 mM HEPES is included in the pipette, and this is critical to the conclusion that pH_i_ increase doesn't have much effect on V_m_. Finally, the numbers of cells (n = 3, 4) are too small to draw statistically significant conclusions, especially when measuring V_m_ of primary cells.

The native I_Ksper_'s Ca^2+^ sensitivity is not that strong, at least not until [Ca^2+^]_i_ reaches 100 μM (Figure 3). Based on Figure 6, the authors conclude that physiologically stimulated [Ca^2+^] increases can reach the levels sufficient to activate I_Ksper_. The Ca^2+^ measurements in Figure 6 with high concentration of progesterone (2 μM) do not have calibration (though with indicators of various Ca^2+^ affinity) and, more importantly, are from whole sperm. The majority of the signal is likely from sperm head where Slo3 is not localized (Figure 1). Therefore, it's hard to make meaningful correlation between the Progesterone-induced Ca^2+^ increase and _IKsper_ activation.

[Editors' note: further revisions were requested prior to acceptance, as described below.]

Brenker et al. report that the principal human K channel, like that in mouse, is encoded by Slo3. This contrasts with the recent *eLife* paper by Mannowetz et al. that reported that Slo1 was responsible for human KSper. Although the only definitive proof will be to identify a human male with a Slo3 mutation that affects his sperm motility (just as Lishko did for CatSper), the current Brenker data are convincing that human KSper is most likely Slo3.

Unlike mouse KSper, the principal K channel in human sperm is primarily activated by internal Ca^2+^, but only modestly by alkaline pH. This agrees with Mannowetz et al. The primary disparity between the papers is that Mannowetz et al found that the specific Slo1 blocker, Iberiotoxin, blocked human KSper, while Brenker et al find that it does not. The experimental solutions used in the two studies may underlie this difference.

There are many more data in the current paper characterizing human KSper and this will be of interest to the field. The work is well done and we find no fault with the data. The most interesting new data is that human Slo3 differs from mouse through the Slo3 pore-containing subunit, rather than through association with an LRRC52 accessory subunit.

The work presents another interesting example that shows that species-specific regulation might have evolved on this essential ion channel protein in sperm, analogous to the different activation mechanisms previously shown in mouse and human CatSper.

The following issues need to be addressed in the revision: 1) The authors presume that the rapid evolution of gamete genes results in gating ring mutations that confer these changes to human Slo3. It would be of interest to readers to show alignment of mouse, intervening species, and humans to narrow in on the changes most likely to mediate these properties. We do not suggest that the authors sort out the changes by mutagenesis, as this would be a separate work in itself and may take several years to complete.

2) The mass spec identification supporting the conclusion is not shown. To support the findings, indicators of merit (such as number of peptides and protein coverage etc.) for all the proteins identified and mentioned in the text should be provided as a Table in Supplementary Information.

3) Although we understand the reasons for using 1 mM Ca^2+^ in some internal solutions to maximize the response, many readers will question the validity of this condition, and it will likely be a source of contention. This should be at least addressed fully in the discussion. Better would be to show data at intervening 10, 30, 100 µM Ca^2+^. However, the reviewers are not worried by the use of 1 mM Ca^2+^. The inclusion of data with other concentrations is optional.

4) Introduction: Navarro et al definitively identified P2X2 channels, not P2X1, in mouse sperm through KO studies. The Brenker study does not identify the P2X channel subtype. After writing a whole paper on the mis-identification of KSper, it is not reassuring that you misidentify P2X in your own summary!

---

## [Author Response]

[Editors’ note: the author responses to the first round of peer review follow.]

While our manuscript was under review, a competing manuscript by [27] entitled “Slo1 is the principal potassium channel of human spermatozoa” appeared in *eLife*. Each manuscript reaches entirely different conclusions regarding the molecular identity of the K^+^ channel underlying I_KSper_ current in human sperm: we conclude that it is Slo3, whereas Mannowetz et al. propose Slo1; both reports cannot be right at the same time.

The two groups agree that the primary human sperm K^+^ current is activated by Ca^2+^.

The major points of difference are: (1) we observe that the human sperm K^+^ current also retains some, albeit modest pH dependence, consistent with the modest pH dependence of heterologously expressed hSlo3; and (2) conflicting pharmacological and biochemical results concerning the molecular identity of the channel.

Concerning the pharmacology, we show that Slo3 inhibitors, but not Slo1 inhibitors, suppress I_KSper_ and depolarize human sperm. In contrast, Mannowetz et al. report that Slo1 inhibitors suppress I_KSper_. Although we do not know the reason for this discrepancy, there are differences in experimental conditions, which we mention in the Discussion of our revised manuscript. Moreover, to ascertain that human Slo3 is insensitive to Slo1 inhibitors, we studied the action of IBTX, CTX, and paxilline on heterologous hSlo3 + LRRC52 channels, see Figure 10 below). We find that human Slo3 is insensitive to IBTX, CTX, and paxilline. As positive control, we used the inhibition of heterologous hSlo1 by IBTX.Author response image 1.Relative amplitude of hSlo1 (at 80 mV) before and after superfusion with IBTX (100 nM). Relative amplitude of hSlo3 + hLRRC52 currents (at 80 mV) before and after superfusion with IBTX (100 nM), CTX (1 μM), or Paxilline (100 nM). hSlo1 and hSlo3 + hLRRC52 were expressed in CHO cells.

In our revised manuscript, we include additional data, showing that progesterone inhibits hSlo3 with a K_i_ of 17 μM, consistent with the inhibition of I_KSper_ in human sperm (27). We further show that Slo1 is largely insensitive to progesterone. Moreover, we show single-channel recordings from human sperm that reveal a Ca^2+^-activated K^+^ channel of ∼ 60-70 pS conductance, i.e., similar to the conductance of human Slo3, but inconsistent with a Slo1 conductance of ∼ 240 pS. These new results further strengthen our conclusion that Slo3 underlies I_KSper_ in human sperm, but are inconsistent with Slo1 properties.

To implement the new data and to improve readability, we rearranged Figures 1, 2, 3, 4 and 5 and we include two new Figures.

Reviewer 1:

*The authors present compelling evidence that that Ksper current in human sperm is carried by the Slo3 potassium channel, as in mouse sperm, but the channel is gated by calcium, unlike in mouse where it is gated by protons. Heterologous expression studies confirm that human Slo3 seems to be calcium gated*.

*The results suggest that the Slo3 channel is gated by different ligands in different species. It is puzzling that the authors agree with the results of the Lishko lab that human Ksper is activated by calcium, but disagree as to which channel it is, as the Lishko paper argues that the mouse Ksper is in fact the (normally calcium gated) Slo1. I find no reason to doubt either study's conclusions and I expect the controversy to be settled with future work*.

*I believe the paper to be appropriate for publication in* eLife, *especially as the Lishko paper appeared there also*.

We thank the referee for the positive comments on our manuscript.

Reviewer 2:

*However, mechanistically and conceptually the advances of the current manuscript are limited. The salient role of Slo3 in mouse sperm has recently been highlighted by several groups. It is well documented in the literature that mouse Slo3 is rather insensitive to calcium and that human Slo3 coexpressed with LRRC52 is less sensitive to alkaline pH as compared to mouse. The authors confirm these observation by others by direct electrophysiological recordings in human sperm. However, no information is provided on the molecular basis underlying these functional differences. Thus, the present manuscript still leaves central mechanistic questions unanswered and the paper is confirmatory to quite some extent*.

The referee’s impression seems to be that the main purpose of our manuscript was to establish that the K^+^ current in human sperm is regulated by pH, like mouse KSper and Slo3. If that were the case, we would certainly agree that the manuscript contributes little new information. However, the focus of the manuscript is in establishing that the human sperm K^+^ current is, in fact, regulated by Ca^2+^, with weaker regulation by pH. The prototypical Ca^2+^-activated and pH-activated members of this channel family are Slo1 and Slo3, respectively. Thus, the important question is whether the human sperm K^+^ current is carried by Slo1 or Slo3. Our results demonstrate that human I_KSper_ current arises from Slo3 and not Slo1. Furthermore, we show that heterologously expressed human Slo3 is Ca^2+^ regulated. These observations are entirely new and it is surprising that two orthologous proteins, mSlo3 and hSlo3, have such different regulation by cytosolic ligands.

Identification of the site on the hSlo3 gating ring underlying regulation by Ca^2+^ and H^+^ is highly interesting, but is beyond the scope of this manuscript. However, we acknowledge that the extensive introductory paragraph on Slo gating rings in our original manuscript might have led the referee astray. Therefore, we shortened this paragraph and moved it to the Discussion.

Reviewer 3:

This referee raised several concerns about the reliability of our electrical recordings from human sperm, in particular, the activation of I_KSper_ by Ca^2+^ and pH_i_. This is somewhat surprising given that the referee might have also seen the Mannowetz et al. manuscript; at least in the case of Ca^2+^, this is not a point of contention between the two groups. Moreover, by manifold controls, we scrutinized the reliability of our recordings:

To ascertain that the voltage-, pH_i_-, and Ca^2+^-activated currents are carried by K^+^ channels, we determined the reversal potential V_rev_ of currents at high and low [K^+^]_o_ and [Cl^-^]_o_.To distinguish experimentally I_KSper_ from monovalent _ICatSper_, we substituted intracellular K^+^ by Cs^+^ and used 1 mM Ca^2+^ in the extracellular medium.We show activation of I_KSper_ by alkalization both by superfusion with NH_4_Cl and by using different pH values in the pipette. We studied I_Ksper_ at low and high [Ca^2+^]_i_ in one and the same cell by photorelease of Ca^2+^ from caged Ca^2+^, while monitoring [Ca^2+^]_i_ with a fluorescent indicator.Ca^2+^ activation of I_KSper_ was studied at 6 different intracellular Ca^2+^ concentrations, ranging from 2.5 to 1000 μM.pH_i_- and Ca^2+^ activation of I_KSper_ was confirmed by current-clamp experiments.We tested Slo1 inhibitors both in voltage- and current clamp.We tested Slo3 inhibitors both in voltage- and current clamp.We studied both Slo1 and Slo3 expression in human sperm by targeted mass spectrometry.Finally, heterologous expression of Slo3 in both oocytes and CHO cells corroborated our electrophysiological and pharmacological conclusions drawn from sperm recordings.

Altogether, the identification of the hSlo3 protein in human sperm by two independent techniques, the pharmacological fingerprint of human I_KSper_ and hSlo3, and the Ca^2+^-activation of heterologous hSlo3 both in oocytes and CHO cells provide compelling support for our conclusion that the Ca^2+^-activated I_KSper_ in human sperm arises from hSlo3, but not Slo1.

*There are three major conclusions in this paper. First, the human K*^*+*^
*current I*_*Ksper*_
*is activated by intracellular Ca*^*2+*^*, rather than pH increase as previously established in mice. Second, Ca*^*2+*^
*increase in human sperm under “physiological” conditions (presumably through Catsper) is sufficient to activate I*_*Ksper*_*. Third, heterologously expressed human Slo3 (hSlo3) is activated by high [Ca*^*2+*^*]*_*i*_*, but is less sensitive to pH. These findings, if confirmed, represent a major step forward in our understanding of sperm membrane potential regulation and fertilization. The third conclusion is very well supported by careful and well-controlled analysis using heterologous expression in both Xenopus oocytes and CHO cells, with or without the auxiliary subunit LRRC52. I find that the first two conclusions, though quite interesting, are not convincing at their present form*.

We provide plenty of experimental evidence that Ca^2+^ activates I_KSper_ (see our general comment above). Therefore, we disagree with this statement. The issue concerning the activation of I_KSper_ by physiological Ca^2+^ concentrations is addressed in more detail below.

*The major data supporting the conclusion that human I*_*Ksper*_
*is potentiated by Ca*^*2+*^
*but not pH*_*i*_
*is in*
Figures 2 and 3.

There seems to be a misunderstanding. We do not claim that human I_KSper_ is exclusively controlled by Ca^2+^. In fact, we show that at pipette pH 6.2, alkalization by NH_4_Cl enhances currents by about 2-fold. Similarly, increasing pipette pH from 6.2 to 7.3 enhances human _IKSper_ by about 2-fold.

*In*
Figure 2
*where the authors want to show what Ca*^*2+*^*, not pH, controls V*_*m*_
*in human sperm, two [Ca*^*2+*^*]*_*i*_
*conditions are compared: 0 Ca*^*2+*^
*and 1 mM Ca*^*2+*^*. The 0 Ca*^*2+*^
*pipette had 1 mM EGTA with no added Ca*^*2+*^*, which results a free [Ca*^*2+*^*] of perhaps < 10 nM. The pipette solution contains no ATP or GTP. Such condition may be fine for studying the biophysical properties of a channel with voltage-clamping, but is too “unphysiological” for studying the regulation of membrane potential. It is hard to interpret the apparent lack of large effect on V*_*m*_
*by pH*_*i*_
*change under this condition. The authors also cited the paper by Navarro (PNAS 104:7688-7692) to compare the difference between human and mouse sperm in the effects of pH*_*i*_
*on V*_*m*_*. However, there is large difference between the pipette solutions used in the two studies (Navarro et al. included ATP and GTP in the pipette and had apparently different free Ca*^*2+*^
*concentration). The authors should at least test several Ca*^*2+*^
*concentrations close to the physiological levels*.

Both Ca^2+^ and pH do affect human I_KSper_ and alkalization does hyperpolarize sperm; yet, Ca^2+^ evokes a more pronounced hyperpolarization. To make comparison of current-clamp and voltage-clamp experiments meaningful, we used the same intracellular solution (without ATP/GTP) throughout the manuscript. Of note, Mannowetz et al. used a similar intracellular solution (also without ATP/GTP) to study human I_KSper_. Under these conditions, our current clamp and voltage-clamp data are consistent, e.g., we observe modest pH_i_ control, but stronger Ca^2+^ control of _IKSper_ and V_m_. Current-clamp experiments are always to a certain degree nonphysiological, no matter whether intracellular ATP/GTP is present or not: the cell’s cytoplasm is exchanged by artificial intracellular solutions and, moreover, experiments are hampered by the high input resistance of sperm (discussed in Zheng et al. 2013, JGP). Therefore, we have tempered our assertions that Ca^2+^ rather than pH_i_ controls V_m_ of human sperm to our recording conditions. Moreover, we now include a paragraph in the discussion stating that it will be an interesting question for future work to determine the relative contributions of cytosolic pH and [Ca^2+^] to the regulation of human sperm function.

In the current-clamp experiments, we used a pipette [Ca^2+^] of either a few nanomolar or 1 mM to study the action of Ca^2+^. As requested, we now include current clamp data at 1, 30, and 70 μM Ca^2+^, showing that the effect of 1 μM Ca^2+^ is miniscule, whereas 30 μM evokes a sizeable hyperpolarization that saturates at ≤ 70 μM. Thus, the concentration dependence of Ca^2+^ action is similar in current- and voltage-clamp experiments.

*In addition, all the pipette solutions have 50 mM HEPES. It's not clear whether 10 mM NH*_*4*_*Cl in the bath is able to significantly increase pH*_*i*_
*when 50 mM HEPES is included in the pipette, and this is critical to the conclusion that pH*_*i*_
*increase doesn't have much effect on V*_*m*_.

In fact, 10 mM NH_4_Cl increases pH_i_ by about 1 unit under the conditions used (see response to Reviewer 2). The modest action of NH_4_Cl on I_KSper_ and V_m_ is consistent with the modest changes of I_KSper_ and V_m_ observed when the pipette pH was changed from pH 6.2 to 7.3. Moreover, at 50 mM intracellular HEPES, we routinely observe NH_4_Cl-induced CatSper activation (see for example [46], [8]). Altogether, these results show unequivocally that the modest action of pH_i_ on I_KSper_ and V_m_ is not due to the specifics of our experimental conditions.

*Finally, the numbers of cells (n = 3,4) are too small to draw statistically significant conclusions, especially when measuring V*_*m*_
*of primary cells*.

We take issue with this comment. First, the small s.d. shows that the V_m_ values are highly reproducible. Second, the current-clamp and voltage-clamp results are consistent with each other in all aspects. Third, we tackled each issue by at least 2 different experimental approaches, e.g., we study Ca^2+^ activation of I_KSper_ by both photorelease of Ca^2+^ and at different Ca^2+^ concentrations in the pipette; we examine pH_i_ activation of I_KSper_ by both superfusion with NH_4_Cl and by different pipette pH; we study the pharmacology of Slo3 and Slo1 inhibitors in voltage- and current clamp; we study Slo3 currents both in oocytes and CHO cells. These different approaches all lead to the same conclusions. Finally, we note that for most experiments n≥4.

*The native I*_*Ksper*_*'s Ca*^*2+*^
*sensitivity is not that strong, at least not until [Ca*^*2+*^*]*_*i*_
*reaches 100 μM (*Figure 3*). Based on*
Figure 6*, the authors conclude that physiologically stimulated [Ca*^*2+*^*] increases can reach the levels sufficient to activate I*_*Ksper*_*. The Ca*^*2+*^
*measurements in*
Figure 6
*with high concentration of progesterone (2 μM) do not have calibration (though with indicators of various Ca*^*2+*^
*affinity) and, more importantly, are from whole sperm. The majority of the signal is likely from sperm head where Slo3 is not localized (*Figure 1*). Therefore, it's hard to make meaningful correlation between the Progesterone-induced Ca*^*2+*^
*increase and I*_*Ksper*_
*activation.*

Figure 3 (now also Figure 2) and Figure 6 (former Figure 5, heterologous expression of Slo3) show that [Ca^2+^] (not 100 μM as this referee states) enhances both I_KSper_ and hSlo3 currents. We agree with the referee on potential limitations of the Ca^2+^ indicator experiments in terms of defining absolute concentrations. However, although Figure 8 (the former Figure 6) does not show calibrated changes in [Ca^2+^]_i_, it clearly argues (in particular the responses recorded with the low-affinity indicator Fluo-5N, K_d_ = 90 μM) that [Ca^2+^]_i_ increases globally μM. In fact, it has been shown that progesterone-induced Ca^2+^ signals originate in the flagellum and propagate in a tail-to-head direction (Servin-Vences et al., 2013, Reproduction). Therefore, the local [Ca^2+^] in the flagellum is predicted to be even higher than in the head. However, we state now in the text that the interplay between CatSper and Slo3 requires further experimentation.

[Editors' note: further revisions were requested prior to acceptance, as described below.]

*Unlike mouse KSper, the principal K channel in human sperm is primarily activated by internal Ca*^*2+*^*, but only modestly by alkaline pH. This agrees with Mannowetz et al*.

We trust that the reviewer means that both studies show that human sperm K^+^ current is activated by Ca^2+^. The two papers reach different conclusions regarding the role of alkaline pH. Mannowetz et al. claim that human I_KSper_ is completely insensitive to pH_i_, whereas we and a recent paper by Mansell et al. (Mol. Hum. Reprod. 2014) find that human I_KSper_ is enhanced by alkalization.

*The primary disparity between the papers is that Mannowetz et al found that the specific Slo1 blocker, Iberiotoxin, blocked human KSper, while Brenker et al find that it does not. The experimental solutions used in the two studies may underlie this difference*.

Although the pharmacological differences between the two studies remain difficult to explain, we note that the intracellular and extracellular [Ca^2+^] used in the pharmacological experiments differ. Overall, our pharmacological profile is consistent with the biochemical evidence for Slo3, the single-channel fingerprint, and the progesterone sensitivity of human I_KSper_.

*The following issues need to be addressed in the revision*:

*1) The authors presume that the rapid evolution of gamete genes results in gating ring mutations that confer these changes to human Slo3. It would be of interest to readers to show alignment of mouse, intervening species, and humans to narrow in on the changes most likely to mediate these properties. We do not suggest that the authors sort out the changes by mutagenesis, as this would be a separate work in itself and may take several years to complete*.

We include the requested alignment. Unfortunately, the alignment does not provide insights regarding the molecular determinants underlying the different ligand sensitivities of human versus mouse Slo3. Except for the Ca^2+^ bowl residues, the determinants of the RCK1 high- affinity site and the lower-affinity site involve multiple residues scattered throughout RCK1. For lower-affinity sites, recognition of any “site” by sequence gazing would be remarkable.

*2) The mass spec identification supporting the conclusion is not shown. To support the findings, indicators of merit (such as number of peptides and protein coverage etc.) for all the proteins identified and mentioned in the text should be provided as a Table in Supplementary Information*.

We now include a Supplementary file with all indicators of merit for the mass spectrometric results

*3) Although we understand the reasons for using 1 mM Ca*^*2+*^
*in some internal solutions to maximize the response, many readers will question the validity of this condition, and it will likely be a source of contention. This should be at least addressed fully in the discussion. Better would be to show data at intervening 10, 30, 100 µM Ca*^*2+*^*. However, the reviewers are not worried by the use of 1 mM Ca*^*2+*^*. The inclusion of data with other concentrations is optional*.

Please note that, in Figure 2, we do, in fact, show current clamp data for 1, 30, 70, and 1,000 µM [Ca^2+^]_i_. In Figure 3 and Figure 3—figure supplement 1, we show I-V curves for I_KSper_ at 2.5, 10, 40, 100, and 1,000 µM [Ca^2+^]_i_. Moreover, in Figure 3, we show original current traces at 0, 40, and 1,000 µM [Ca^2+^]_i_. We feel this data should address the concerns of the reviewer.

There are three experiments where only 1 mM [Ca^2+^]_i_ was used:

1) to study the pharmacology of I_KSper_,2) to show that at high [Ca^2+^]_i_, V_m_ is independent of [Cl^-^]_o_,3) to show that at high [Ca^2+^]_i_, I_KSper_ is still potentiated by alkalization.

For experiment 1, we provide the rationale for using 1 mM [Ca^2+^]_i_ in the results section and in the discussion. For experiments 2 and 3, it is also advisable if not required to use high [Ca^2+^]_i_. Accordingly, we do not expect the high concentrations a potential source of contention. Therefore, we prefer to leave the Discussion as it is.

*4) Introduction: Navarro et al definitively identified P2X2 channels, not P2X1, in mouse sperm through KO studies. The Brenker study does not identify the P2X channel subtype. After writing a whole paper on the mis-identification of KSper, it is not reassuring that you misidentify P2X in your own summary*!

Although we assume that the reviewer made this comment in jest, our work was not about mis-identification of KSper, but simply reports our experiments showing that human KSper is regulated by both cytosolic calcium and alkalization, and that properties of heterologously expressed hSlo3 are consistent with human KSper. Moreover, we thank the referee pointing out the typo regarding P2X channels; we have made some clarifying changes to the respective sentence: “purinergic P2X channels are functional in mouse (31), but not in human sperm (8)”.